# Machine learning identifies molecular regulators and therapeutics for targeting SARS-CoV2-induced cytokine release

Marina Chan[1], Siddharth Vijay[1], John McNevin[2] , M Juliana McElrath[2] , Eric C Holland[1,*] &
Taranjit S Gujral[1,3,**]

## Abstract

Although 15–20% of COVID-19 patients experience hyper-inflammation induced by massive cytokine production, cellular triggers of this process and strategies to target them remain poorly understood. Here, we show that the N-terminal domain (NTD) of the SARS-CoV-2 spike protein substantially induces multiple inflammatory molecules in myeloid cells and human PBMCs. Using a combination of phenotypic screening with machine learning-based modeling, we identified and experimentally validated several protein kinases, including JAK1, EPHA7, IRAK1, MAPK12, and MAP3K8, as essential downstream mediators of NTD-induced cytokine production, implicating the role of multiple signaling pathways in cytokine release. Further, we found several FDA-approved drugs, including ponatinib, and cobimetinib as potent inhibitors of the NTD-mediated cytokine release. Treatment with ponatinib outperforms other drugs, including dexamethasone and baricitinib, inhibiting all cytokines in response to the NTD from SARS-CoV-2 and emerging variants. Finally, ponatinib treatment inhibits lipopolysaccharide-mediated cytokine release in myeloid cells *in vitro* and lung inflammation mouse model. Together, we propose that agents targeting multiple kinases required for SARS-CoV-2-mediated cytokine release, such as ponatinib, may represent an attractive therapeutic option for treating moderate to severe COVID-19.

**Keywords** kinases; machine learning; N-terminal domain; Ponatinib; SARS-CoV-2

**Subject Categories** Immunology; Microbiology, Virology & Host Pathogen Interaction; Pharmacology & Drug Discovery

**Mol Syst Biol. (2021) 17: e10426**

## Introduction

The severe acute respiratory syndrome coronavirus 2 (SARS-CoV-2) is responsible for the current coronavirus disease 2019 (COVID-19) pandemic. COVID-19 typically presents with symptoms attributed to viral replication that resolve within 1–2 weeks. In approximately 15–20% of cases, acute infection is followed by more serious events where myeloid cells, including monocytes and macrophages, produce a cytokine storm with the rapid release of IL-6, IL-1b, CXCL10, CCL7, and other inflammatory molecules (Monteleone *et al*, 2020). The appearance of cytokines in patient samples is concomitant with increased viral load, loss of lung function, lung injury, and a fatal outcome (Vaninov, 2020). Drugs or drug combinations that reduce the cytokine storm, therefore, would be helpful in the treatment of COVID-19.

A recent study using a myeloid receptor-focused ectopic expression screen identified several C-type lectins and Tweety family member 2 as glycan-dependent binding partners of the SARS-CoV-2 spike protein (Lu *et al*, 2021). The engagement of these receptors with the SARS-CoV-2 virus induces robust proinflammatory responses in myeloid cells that correlate with COVID-19 severity. The downstream signaling pathways that induce severe presentation, however, are not fully identified, impeding the development of targeted therapies. Here, we combined approaches from immunology, systems biology, and biochemistry to uncover underlying kinase-driven signaling networks triggering myeloid cell cytokine release. We identified and validated FDA-approved, or clinical-grade compounds inhibiting inflammatory cytokine production in the context of SARS-CoV-2. Our goal was to determine if FDA-approved drugs available for clinical use might forestall drug development and deliver a timely solution for the COVID-19 pandemic.

## Results

We used the macrophages derived from immortalized monocyte-like cell line THP1(Bosshart & Heinzelmann, 2016) to model

---

1   Human Biology Division, Fred Hutchinson Cancer Research Center, Seattle, WA, USA
2   Vaccine and Infectious Disease Division, Fred Hutchinson Cancer Research Center, Seattle, WA, USA
3   Department of Pharmacology, University of Washington, Seattle, WA, USA
   *Corresponding author. Tel: +1 206 667 6117; E-mail: eholland@fredhutch.org
   **Corresponding author Tel: +1 206 667 4149; E-mail: tgujral@fredhutch.org

myeloid cell-SARS-CoV-2 interaction *in vitro* and mammalian HEK293 cells that produced full-length Spike subunit S1 protein (Val[16]-Gln[690]); the SARS-CoV-2 protein is essential for host cell entry (Fig 1A). Consistent with previous observations, our preliminary data show that 24-h stimulation with full-length mammalian cell-derived S1 subunit of SARS-CoV-2 spike protein causes massive upregulation of IL-1b in a dose-dependent manner (Fig EV1). We asked if S1 protein could promote upregulated expression of cytokines observed to be elevated in COVID-19 patients. Our data show that S1 protein stimulation (1 μg/ml) causes a significant increase in the expression of a panel of cytokines including IL-8 (~96-fold), IL-6 (5-fold), IL-1b (~120-fold), TNF (~15-fold), and chemokines including CXCL10 (600-fold), CCL2 (~70-fold), and CCL7 (4-fold) in THP1-derived macrophages (Fig 1B). Similar changes in cytokines were also observed in healthy donor PBMCs and Raw264.7 mouse monocytes in response to full-length S1 protein (Fig EV1). Together, these data indicate that interaction between Spike subunit S1 protein and myeloid cells is sufficient to activate these cells.

We sought to determine the region of the S1 subunit that mediates myeloid cell activation. Both SARS-CoV-2 and SARS-CoV recognize the angiotensin-converting enzyme 2 (ACE2) receptor in humans (Shang *et al*, 2020). The receptor-binding domain (RBD) of SARS-CoV-2 spike subunit S1 protein binds to ACE2 (Shang *et al*, 2020), promoting virus entry into cells while the function of the N-terminal domain (NTD) is not well understood. Surprisingly, we found that stimulation with Val[16]-Ser[305] NTD of S1 subunit is sufficient to promote cytokine expression while stimulation with Arg[319]-Phe[541] RBD of S1 protein did not activate THP1 cells (Fig 1C). Similar changes in the release of cytokines in the conditioned media were also observed in PBMCs from 20 healthy individuals in response to stimulation with NTD of S1 subunit (Fig 1D). Consistently, treatment with CV30 (Seydoux *et al*, 2020) (a highly potent antibody targeting RBD of SARS-CoV-2) did not change S1 protein-induced expression of cytokines in PBMCs (Fig 1E). Further, stimulation with spike protein from two different endemic coronaviruses (HCoV-COV-2 and HCoV-OC43) did not promote cytokine release in THP1 cells (Fig EV1), suggesting that the S1 subunit of SARS-CoV-2 contains a unique feature of this specific spike protein sequence required for myeloid cell activation. Together, these data indicate that NTD of S1 subunit interacts with a different receptor than ACE2, on myeloid cells to activate "cytokine release".

To identify S1 spike protein-induced signaling pathways activating myeloid cells and to explore potential targets for therapeutic development, we used a recently developed strategy combining phenotypic screening with machine learning-based functional screening approaches, called KiDNN (Vijay & Gujral, 2020) and KiR (Gujral *et al*, 2014b). KiR is based on linear, elastic net regularizations, while KiDNN utilizes nonlinear deep neural networks (DNN). Previously, we showed that KiR uses large-scale drug-target profiling efforts, elastic net regularization, and broadly selective chemical tool compounds to pinpoint specific nodes (kinases and associated networks) underlying a given phenotype, such as cell growth or release of secreted factors, e.g., cytokines (Gujral *et al*, 2014b) (Fig 2 A). In contrast, KiDNN uses DNN to predict responses to inhibitors (Vijay & Gujral, 2020). We screened a set of 35 computationally chosen kinase inhibitors (Gujral *et al*, 2014c) and quantified their effect on changes in NTD-mediated release of seven cytokines in the conditioned media from pooled PBMCs (Fig 2B, Dataset EV1). Using

results from this cytokine release experiment as a training dataset, we built both KiR (Fig EV2) and KiDNN models (Fig EV3) to predict kinases essential for the NTD-mediated release of cytokines. Model performance was evaluated using leave-one-out cross-validation (LOOCV) mean squared error (MSE) between predicted and observed drug response. In LOOCV, each time 34 drugs' activity profiles were used to train the model to predict the remaining drug's effect on NTD-mediated cytokine release. MSE between predicted and observed cytokine levels was used to assign an error score to each model. Overall, models for each of the cytokines performed with at least 85% accuracy (Fig EV2). The optimized models with the least mean squared errors collectively identified 30 most "informative kinases" (out of > 300 kinases) that may be involved in the NTD-mediated cytokine release (Fig EV4). These include several kinases known to play a critical role in cytokine signaling, such as JAK1 and IRAK1, as well as kinases not previously known to play a role in this process, like EPHA7, MAP3K8, and MAP3K2. Overall, MAP3K8 was shown to be enriched in the cytokine-mediated signaling network of all seven cytokines, while EPHA7 was enriched in networks of 4 out of 7 cytokines (Fig EV4). Therefore, based on this analysis, we predicted that both MAP3K8 and EPHA7 are essential for the NTD-mediated cytokine release.

To validate the role of the kinases that we predicted to be important in cytokine signaling, we examined the effects of depleting these kinases in gene knockdown experiments. Using a pooled set of four siRNA, we knocked down the expression of top 13 kinases in THP1 cells implicated by KiR analyses (Fig EV4) and measured their effects on NTD-mediated cytokine release. Transient transfections of pooled siRNA led to a 35–80% knockdown of each of the kinases measured by quantitative PCR (Fig EV4). Our data showed that knockdown of 11 out of 13 kinases led to an > 1.5-fold decrease in NTD-mediated cytokine levels compared with scrambled siRNA controls (Fig 2C). Of these, knockdown of MAPK12, EPHA7, MAP3K8, PRKACG, IRAK1, MAP3K3, and JAK1 led to a decrease in more than one cytokine or chemokine (Fig 2C). Network analyses based on prior information showed that several of these kinases are known to regulate common transcription factors such as *MYC* and *REL* (Fig EV4). Together, these data confirmed KiR models' predictions, validating the role of at least seven kinases in the SARS-CoV2 S1 subunit NTD-mediated cytokine release.

Given that our analyses implicated multiple kinases affecting different signaling pathways in triggering cytokine release, we hypothesized that a rational drug combination, or multi-targeted therapy, may represent an effective approach to blocking the SARS-CoV-2-mediated cytokine storm. Next, we used the optimized KiDNN models to predict both pairwise and single agent responses to 427 single inhibitors that reduce NTD-mediated cytokine levels (Fig 3A). To predict the effect of drug combinations, a linear combination of the activities corresponding to each drug for each kinase was applied to create a pseudo-activity matrix. Subsequently, the pseudo-matrix for all 428 by 428 combinations of drugs was computed and inputted into KiDNN models for prediction. Combinations containing the most promiscuous drugs, such as staurosporine and K252a, were excluded from this analysis. In our analyses, we prioritized those compounds that are FDA-approved for human use, known to exhibit a low toxicity profile, and predicted inhibition of release for multiple cytokines. Our data showed that ponatinib, an FDA-approved drug for chronic myelogenous leukemia, was

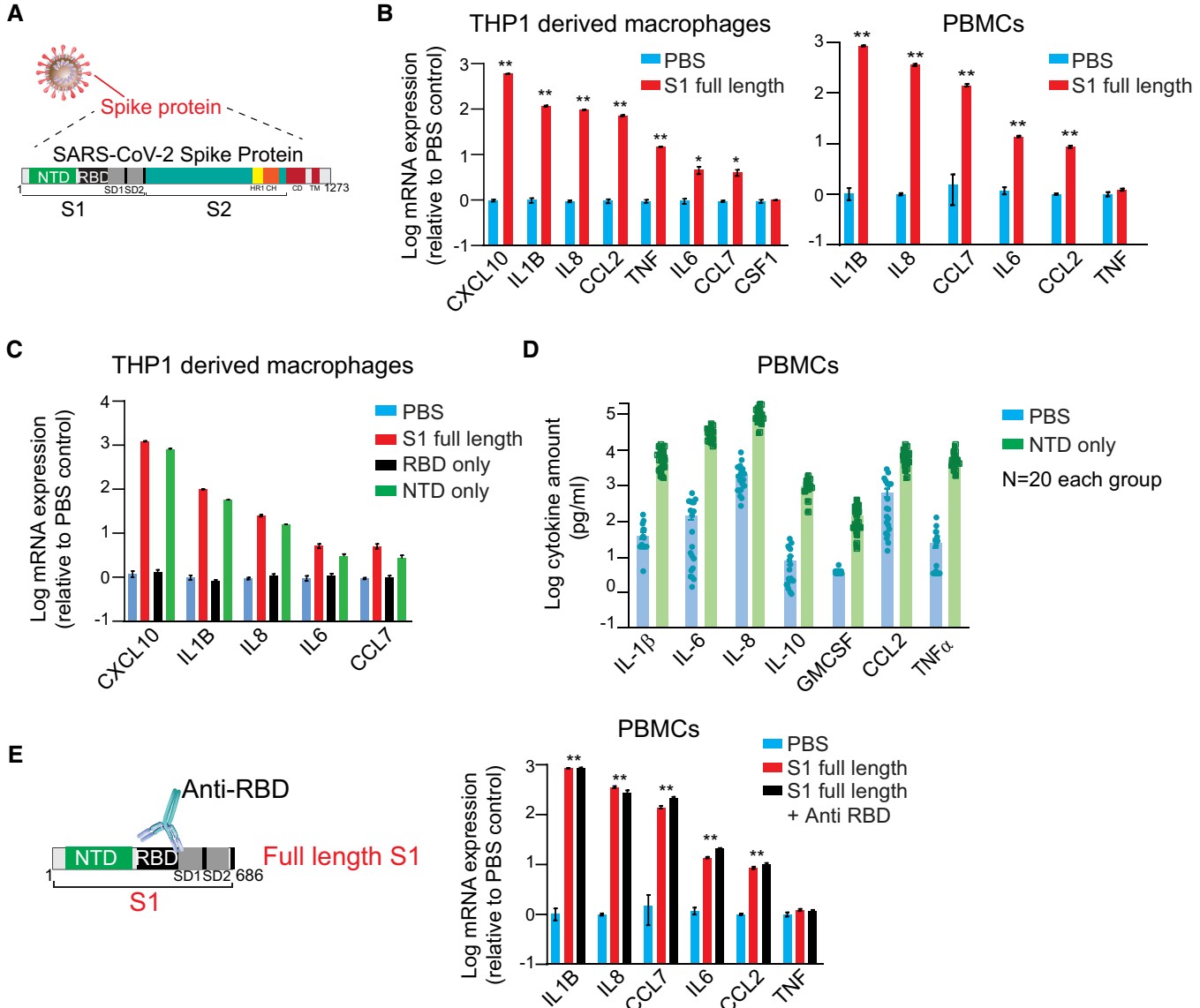

**Figure 1. SARS-Cov-2 Spike subunit S1 protein causes a significant increase in the expression and release of a panel of cytokines in THP1 cells and human PBMCs.**

A    A schematic showing major SARS-CoV-2 proteins and domain structure of spike protein.

B    Changes in the expression of cytokines in THP1 macrophages (left) and PBMCs (right) upon treatment with full-length S1 subunit at 1 μg/ml for 24 h.

C    Changes in the expression of cytokines in THP1 macrophages upon treatment with different domains of S1 subunit at 1 μg/ml for 24 h.

D    Measurement of cytokine release from healthy donor PBMCs treated with PBS or NTD at 1 μg/ml for 24 h.

E    Effect of an anti-RBD antibody on S1 subunit stimulated changes in the expression of cytokines in PBMCs. *Left*, a schematic showing the domain structure of S1 spike protein and anti-RBD antibody. *Right*, changes in cytokine gene expression in response to S1 spike protein in the presence or absence of anti-RBD antibody. Gene expression was measured by qPCR. Cytokine release in the conditioned media was measured by Luminex. Full-length S1 and S1 subunits are purified from HEK293 cells.

Data information, in (B, C, E), data are shown as the mean of three technical replicates; in (D), data are shown as the mean of 20 individual donors. Error bars denote SEM. *$P < 0.05$, **$P < 0.01$, Multiple *t*-test.

predicted as the most effective in blocking all seven cytokines as a single agent, as well as in combination (top 360 out of 500 combinations). However, a combination of other drugs with ponatinib marginally improved overall response, indicating the effect of drug combination is primarily driven by ponatinib. We experimentally

tested the efficacy of 10 compounds *in vitro*, including 5 FDA-approved drugs, that we predicted to inhibit NTD-mediated cytokine release (Fig 3B). Treatment with ponatinib and cobimetinib potently inhibits NTD-mediated cytokine release in PBMCs. In addition, treatment with sunitinib and bosutinib also inhibited all seven cytokines,

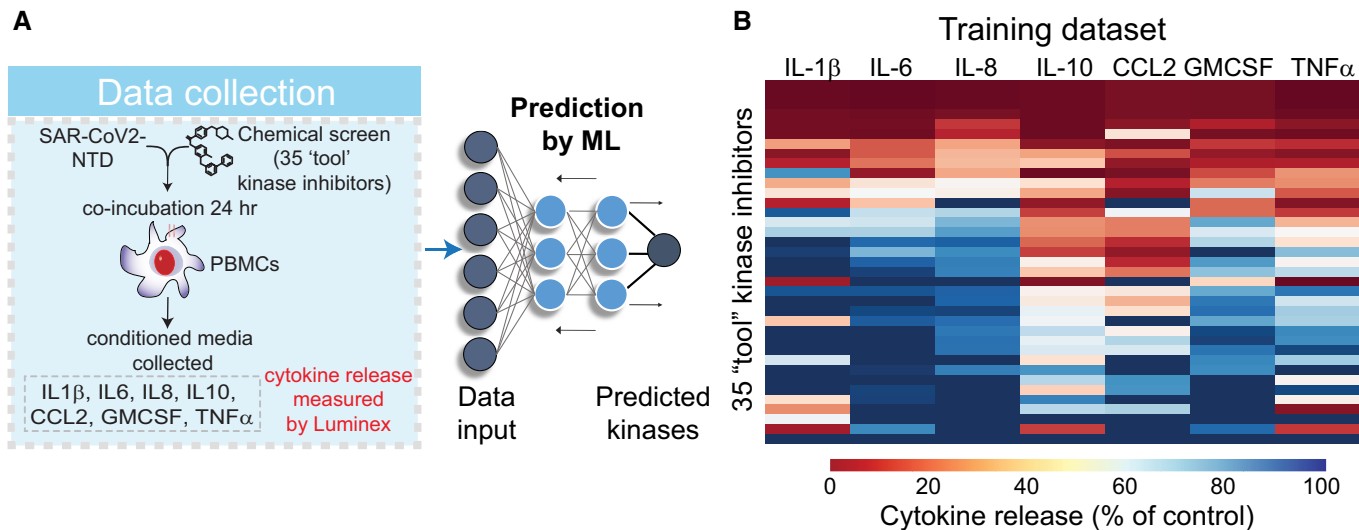

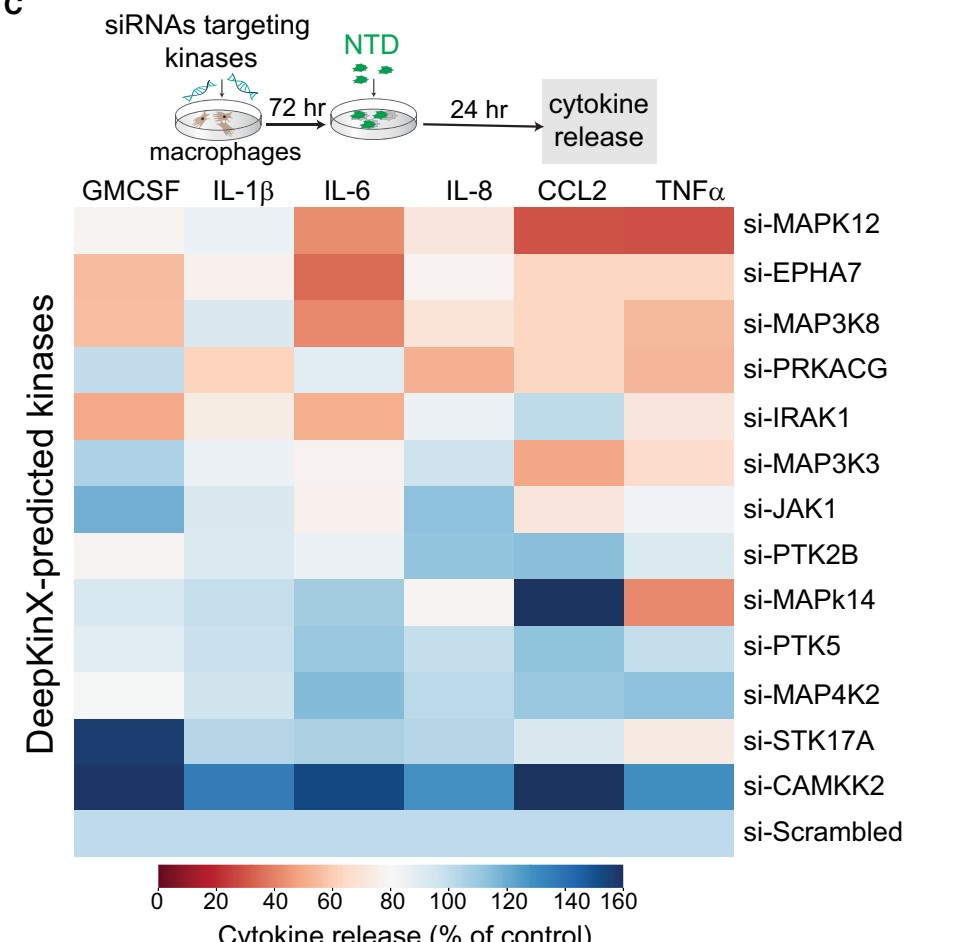

**Figure 2. Machine learning-based functional screening identified key kinase drivers of the NTD-stimulated cytokine release in PBMCs.**

A  A schematic showing the process of the machine learning-based functional screening.

B  A heatmap showing changes in the indicated cytokines in response to treatment of PBMCs with 35 "tool" kinase inhibitors at 500 nM as the training set. Cytokines released in the media were measured by Luminex and normalized to DMSO control.

C  Validation of the predicted kinases as drivers of cytokine release by siRNAs in THP1 cells. Cytokine release was measured by Luminex and normalized to scrambled siRNA.

although to a lesser extent (Fig 3B). We also validated the response to axitinib, which was predicted to be ineffective by our model (Dataset EV1). Importantly, treatment with ponatinib outperforms baricitinib, a JAK inhibitor FDA-approved for the treatment of COVID-19 (Favalli *et al*, 2020), in inhibiting all seven cytokines in response to NTD in PBMCs (Fig 3A). Kinase activity profiling of ponatinib showed that this drug inhibits 8 out of 10 experimentally validated kinases necessary for the NTD-mediated cytokine and chemokine release (Fig 3C and D). In contrast, baricitinib was shown to inhibit only 4 of these essential kinases. Thus, our data indicate that ponatinib, a multi-specific kinase inhibitor, blocks the activity of several kinases that are essential for cytokine signaling, thereby dampening the NTD-mediated cytokine production.

Clinical presentation of cytokine storm has been associated with systemic infections such as sepsis and after immunotherapies such as Coley's toxins (Fajgenbaum & June, 2020). Previous studies have shown that a subset of monocytes may be infected by SARS-CoV-2 and differentiate into infected macrophages in local tissue, contributing to the inflammatory cytokine storm in patients (Jafarzadeh *et al*, 2020) (Guo *et al*, 2020). Therefore, we asked whether PBMCs isolated from COVID-19 patients respond differently to NTD treatment from healthy donor PBMCs. We found that the fold change in a subset of cytokines released in response to NTD stimulation was significantly higher in COVID-19 PBMCs than healthy donor PBMCs (Fig EV5). Whether this difference can be attributed to the duration of infection, clinical presentation, or patient characteristics such as age and pre-existing conditions warrants future investigations. Nonetheless, ponatinib still outperforms dexamethasone in inhibiting NTD-stimulated cytokine release from COVID-19 PBMCs (Fig EV5).

Recently, several new variants of SARS-CoV-2 have emerged, including the B.1.1.7 first identified in the United Kingdom, B.1.351 identified in South Africa, and more recently B.1.165.2 (delta variant), that harbor mutation sites in the NTD. Therefore, we asked if these variants could also promote cytokine release. We show that all three B1.1.7, B.1.351, and B.1.165.2 NTDs could promote the release of cytokines in PBMCs at the levels comparable with the NTD from Wuhan SARS-CoV-2 (Fig 4A). Notably, ponatinib treatment could inhibit the NTD-mediated release of cytokines from all variants. Ponatinib also outperforms both dexamethasone and baricitinib to block the cytokine release, including IFNγ, even at a substantially reduced dose (125 nM ponatinib vs. 1,000 nM baricitinib or dexamethasone) (Figs 4A and EV6). Accordingly, we found the effective ponatinib concentration range for the panel of cytokines tested, required to inhibit 50% of NTD-mediated cytokine release ($EC_{50}$), was between 3 and 25 nM (Fig 4B), further demonstrating a strong potency of ponatinib in inhibiting cytokine release caused by NTD from SARS-CoV-2 and various emerging variants.

A functional T-cell response is critical to the resolution of viral infection. We therefore employed an *in vitro* Activation-Induced Marker (AIM) assay to assess the effect of ponatinib on T-cell response at concentrations that strongly inhibit myeloid cell-induced cytokine production. *In vitro* AIM assays have been used to determine T-cell response to viral peptides with or without drug interventions, including SARS-CoV-2 (Moderbacher *et al*, 2020). In order to determine whether ponatinib affected anti-viral T-cell function at the concentrations used to block the NTD-induced cytokine release, we stimulated human PBMCs with synthetic cytomegalovirus (CMV) peptide pool in the presence of various concentrations of ponatinib, dexamethasone, and baricitinib. CMV-specific T-cell response was measured as a percentage of AIM+ cells (Fig EV7A). Treatment with ponatinib had little or no effect on CMV peptide-induced $CD8^+$ T-cell activation at 10–40-fold above the $EC_{50}$ concentrations of inhibiting the myeloid cell-induced cytokine production, while some inhibitory effect on $CD4^+$ T cells was observed at high concentrations (Fig EV7B). Notably, both dexamethasone and baricitinib induced stronger or similar inhibition on $CD8^+$ and $CD4^+$ T cells at the same doses. In addition, ponatinib did not affect PBMC cell viability (Fig EV7C). Taken together, these data show that ponatinib is highly effective in blocking NTD-mediated cytokine production by myeloid cells while preserving the ability of a T-cell response.

The lipopolysaccharide (LPS)-induced models of acute lung injury (ALI) and acute respiratory distress syndrome (ARDS) in mice are well-established *in vivo* models to study pulmonary infection. Further, both ALI and ARDS are known to occur in the clinical presentation of severe SARS-CoV-2 disease (Li *et al*, 2020). Thus, we sought to compare S1 protein and LPS-mediated changes in cytokine expression in THP1 cells and determine whether ponatinib could inhibit LPS-mediated cytokine production in these cells. Our data show that both S1 spike protein (1 μg/ml) and LPS (1 μg/ml) stimulation increased the expression of all measured cytokines (Fig 5A). However, some specific differences in cytokine expression were also observed: LPS stimulation caused ~1,000-fold increase in the expression of IL-6 compared with 10-fold increase caused by S1 spike protein stimulation (Fig 5A). Conversely, S1 protein stimulation caused ~1,000-fold change in the expression of CXCL10 compared with a 10-fold increase caused by LPS stimulation. Importantly, ponatinib treatment inhibited LPS-mediated expression of IL-1b, IL-6, IL-8, and TNFα (Fig 5B). Consistent with these *in vitro* data, we show that a 1-h pre-treatment with ponatinib (35 mg/Kg) significantly reduces symptoms of acute lung inflammation in the LPS-induced lung inflammation mouse model (Fig 5D and E). Previously, a similar dose of ponatinib and vehicle control was used in cancer mouse models (O'Hare *et al*, 2009; Gozgit *et al*, 2011). Notably, the mean plasma levels reached ~800 and ~550 nM at 2 and 6 h, respectively, after a single dose of 30 mg/kg ponatinib treatment (O'Hare *et al*, 2009). In our study (Fig 5C), treatment with ponatinib at 35 mg/kg alleviated LPS-induced lung injury, including interstitial and intra-alveolar edema, septal thickening, alveolar collapse, and inflammatory cell infiltration, assessed by histology (Fig 5D). Bronchoalveolar lavage fluid (BALF) collected at 5 h post-LPS treatment showed a significant reduction in GMCSF, IL-6, and TNFα levels measured by Luminex (Fig 5E). Together, these data suggest that ponatinib also blocks LPS-mediated downstream signaling and cytokine production.

## Discussion

Most COVID-19 patients develop mild to moderate symptoms, while 15–20% of patients face hyper-inflammation induced by cytokine production leading to respiratory failure. A significant challenge in targeting immune response is an incomplete understanding of how host cells trigger cytokine release. Here, we report both viral and host-specific molecular mechanisms of how SARS-CoV-2 spike

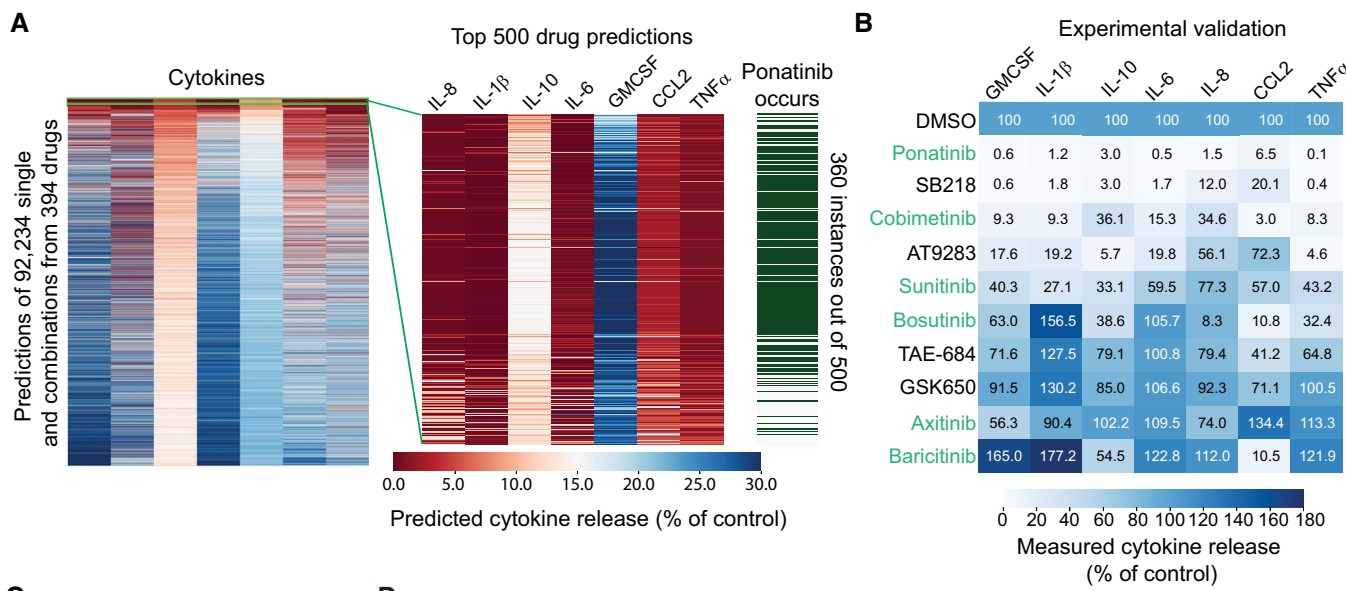

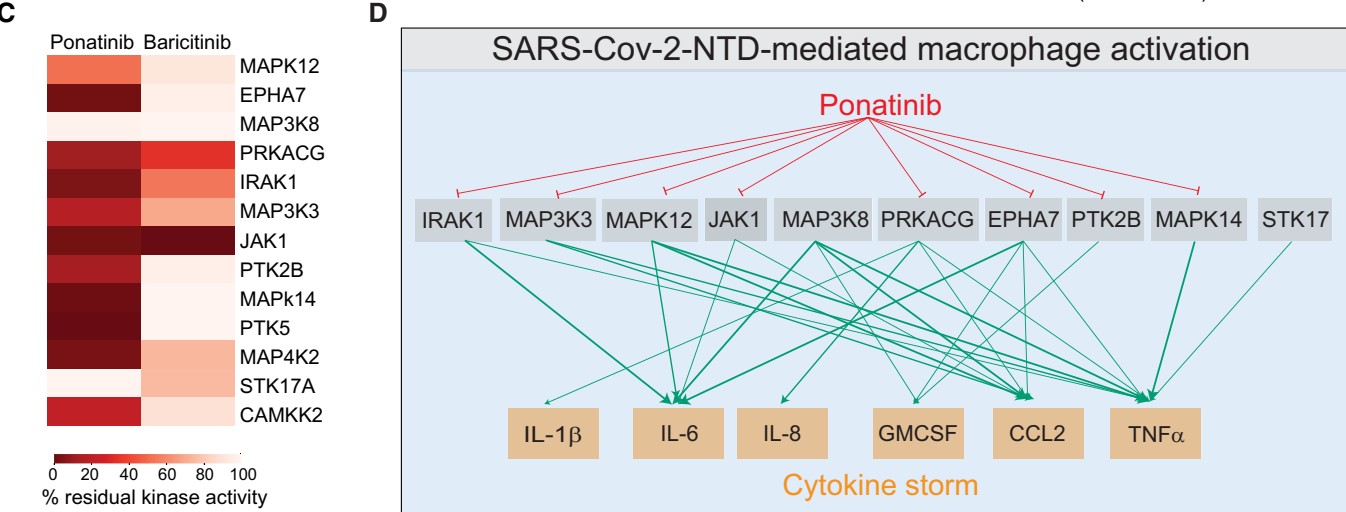

**Figure 3. Machine learning-based functional screening identified ponatinib as a potent inhibitor of the NTD-mediated cytokine release.**

A   A heatmap showing the effect of single and combinatorial drug predictions by machine learning on indicated cytokine release (*left*), a zoomed view of top 500 drug predictions (*middle*), and frequency of ponatinib occurrence in the top 500 predictions (*right*). A total of 92,234 single and combinatorial drug predictions were generated from 394 drugs. Predicted cytokine release is expressed as % of DMSO control.

B   A heatmap showing experimental validation of model-predicted inhibitor response. All inhibitors were tested at 500 nM. Green text indicates an FDA-approved drug. Our model predicted axitinib to be ineffective, therefore, used as a negative control.

C   Comparison of kinase inhibition profile of ponatinib and baricitinib.

D   A schematic showing ponatinib inhibits multiple kinases involved in the SARS-CoV2-NTD-mediated cytokine signaling.

protein induces cytokine release. We discovered a previously unknown function of the NTD of SARS-CoV-2 spike protein in promoting cytokine release in immune cells. Our findings further

highlight the importance of non-RBD region on spike protein and have implication for developing more effective neutralizing antibodies. To identify signaling pathways activated in myeloid cells in

**Figure 4. Ponatinib inhibits the SARS-Cov2 variant NTD-mediated cytokine release.**

A   A heatmap showing changes in indicated cytokines in response to NTD from indicated SARS-Cov-2 variants and inhibitors. Data are shown as fold change between NTD treatment and PBS control. Dex; dexamethasone.

B   Dose-response curves of ponatinib treatment on SARS-Cov2 NTD (Wuhan)-mediated cytokine release in PBMCs. *Right*, a table showing $EC_{50}$ (nM) values of ponatinib-mediated inhibition of cytokine release in response to indicated SARS-Cov2 variant NTD treatments in PBMCs.

▶

**A**

| IL-1β | IL-6 | IL-8 | IL-10 | CCL2 | GM-CSF | TNFα | | |
|---|---|---|---|---|---|---|---|---|
| 1 | 1 | 1 | 1 | 1 | 1 | 1 | PBS | |
| 37 | 52 | 4.6 | 25 | 2 | 40 | 24 | NTD +DMSO | |
| 4.5 | 3.8 | 0.7 | 1 | 0.3 | 1 | 0.2 | NTD + Ponatinib 0.125 μM | Wuhan |
| 10 | 23 | 1.9 | 12 | 2.2 | 4.7 | 6.4 | NTD + Dex 1 μM | |
| 45 | 54 | 3.9 | 9.1 | 0.1 | 61 | 31 | NTD + Baricitinib 1 μM | |
| 18 | 30 | 3.9 | 14 | 2.4 | 21 | 9.6 | NTD + DMSO | |
| 1 | 1.1 | 0.2 | 1 | 0.1 | 1 | 0.1 | NTD + Ponatinib 0.125 μM | U.K. (B.1.1.7) |
| 6.6 | 12 | 1.4 | 5.9 | 2.7 | 2.3 | 2.8 | NTD + Dex 1 μM | |
| 32 | 36 | 3.2 | 6 | 0.1 | 38 | 16 | NTD + Baricitinib 1 μM | |
| 35 | 49 | 4.5 | 24 | 2.2 | 37 | 22 | NTD + DMSO | |
| 3 | 2.4 | 0.4 | 1 | 0.2 | 1 | 0.1 | NTD + Ponatinib 0.125 μM | South Africa (B.1.351) |
| 14 | 21 | 1.8 | 11 | 2.3 | 4.2 | 5.7 | NTD + Dex 1 μM | |
| 40 | 44 | 3.5 | 7.4 | 0.1 | 52 | 27 | NTD + Baricitinib 1 μM | |
| 11 | 20 | 5.8 | 12 | 2.8 | 50 | 5.5 | NTD + DMSO | |
| 1 | 0.9 | 0.2 | 1 | 0.1 | 1 | 0.2 | NTD + Ponatinib 0.125 μM | Delta (B.1.651.2) |
| 3.7 | 7.2 | 1.4 | 3.4 | 1.8 | 4 | 1.5 | NTD + Dex 1 μM | |
| 10 | 22 | 3.5 | 5 | 0.3 | 86 | 3.5 | NTD + Baricitinib 1 μM | |

0   20   40   60   80   100
Fold change in cytokine amount

**B**

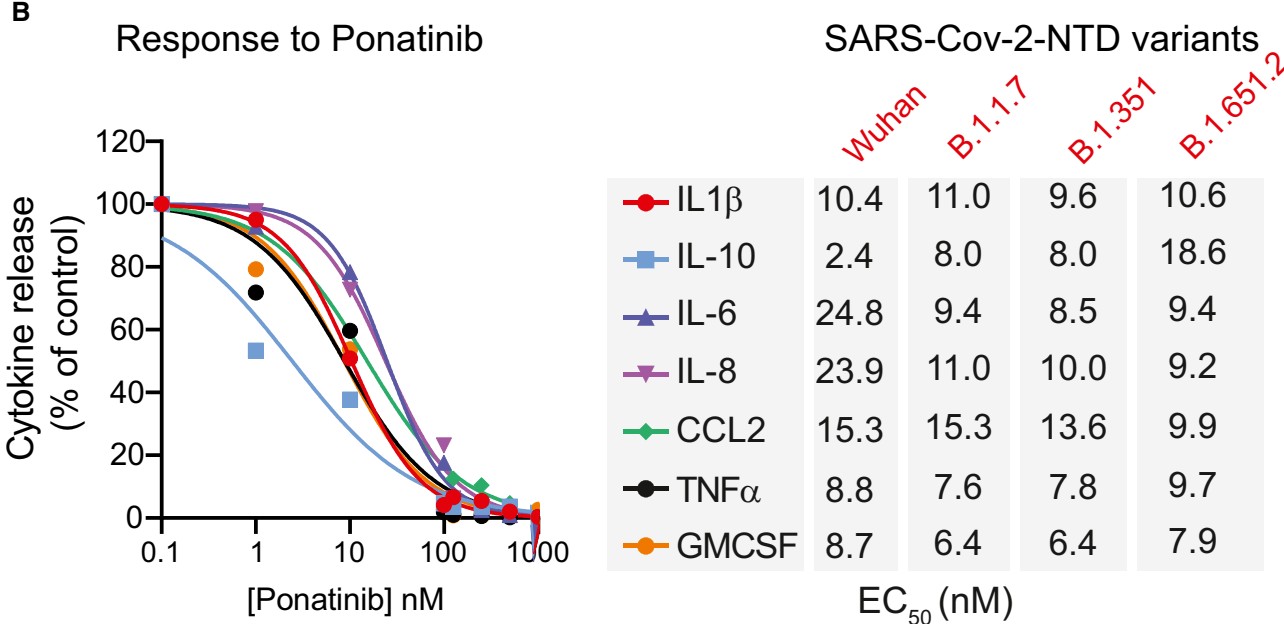

## Response to Ponatinib

## SARS-Cov-2-NTD variants

| | Wuhan | B.1.1.7 | B.1.351 | B.1.651.2 |
|---|---|---|---|---|
| IL1β | 10.4 | 11.0 | 9.6 | 10.6 |
| IL-10 | 2.4 | 8.0 | 8.0 | 18.6 |
| IL-6 | 24.8 | 9.4 | 8.5 | 9.4 |
| IL-8 | 23.9 | 11.0 | 10.0 | 9.2 |
| CCL2 | 15.3 | 15.3 | 13.6 | 9.9 |
| TNFα | 8.8 | 7.6 | 7.8 | 9.7 |
| GMCSF | 8.7 | 6.4 | 6.4 | 7.9 |

$EC_{50}$ (nM)

**Figure 4.**

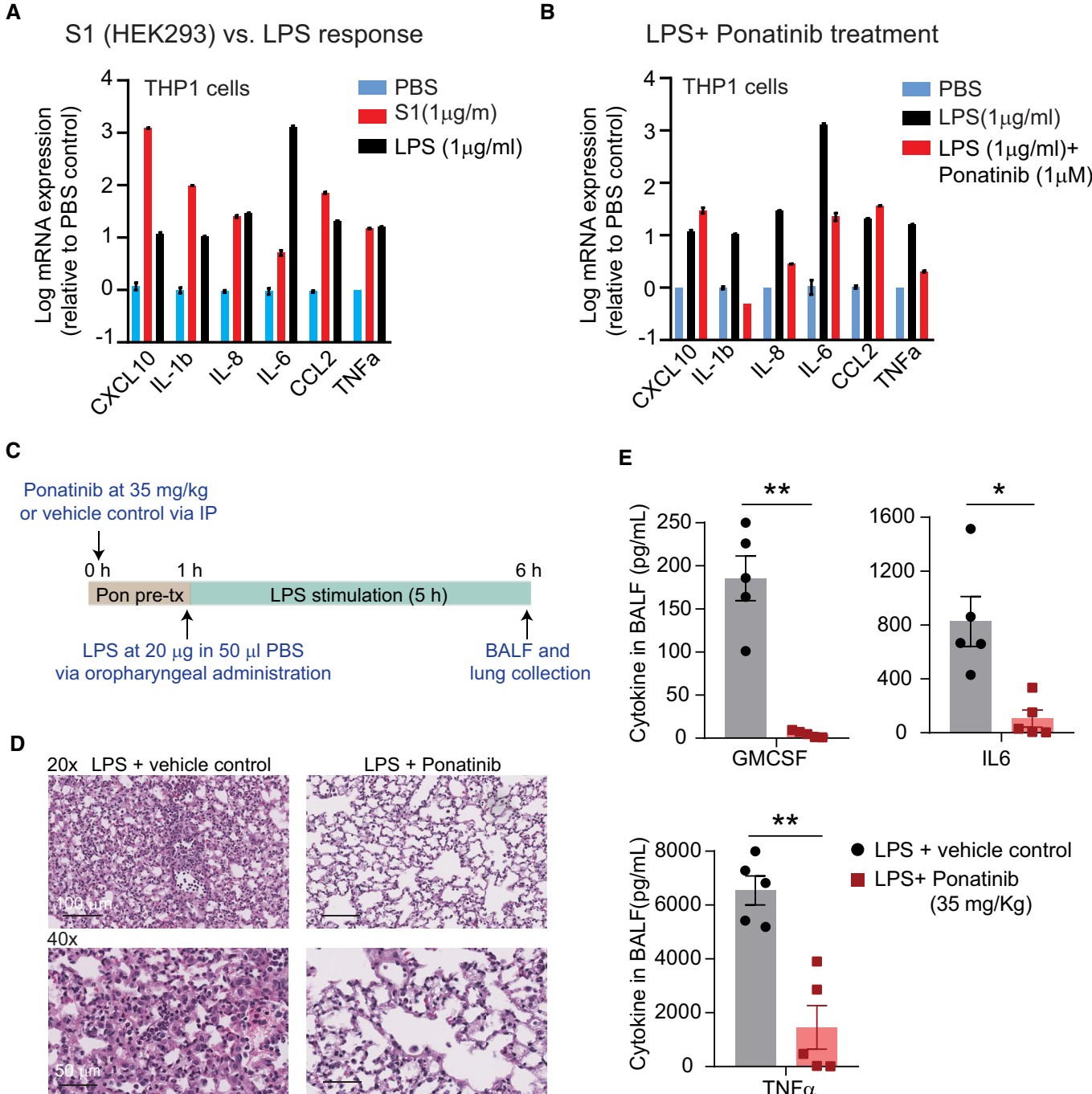

**Figure 5. Ponatinib alleviates symptoms of acute lung inflammation in LPS-induced lung inflammation mouse model.**

A Comparison of LPS and full-length S1 spike protein-mediated changes in cytokines in THP1 macrophages.

B Ponatinib inhibits LPS-mediated cytokine release in THP1 macrophages *in vitro*.

C A schematic showing the overall design of *in vivo* study evaluating the efficacy of ponatinib in LPS-induced lung inflammation mouse model.

D Representative H&E images showing ponatinib alleviates LPS-induced inflammatory cell infiltration, septal thickening, alveolar edema in mouse lungs. Scale bars 100 μm (upper) and 50 μm (lower).

E Plots showing pre-treatment (60 min) with ponatinib (35 mg/kg) inhibits LPS-induced cytokine release in BALF. Cytokine levels were measured using Luminex.

Data information, in (A-B), data are shown as the mean of three technical replicates. Error bars denote SEM. In (E), data are mean of five biological replicates from two independent studies, error bars denote SEM. *$P < 0.05$, **$P < 0.01$, Welch's *t*-test.

response to the NTD of S1 spike protein, we employed a combination of phenotypic screening with machine learning-based modeling. We identified previously known kinases including JAK1 (Luo *et al*, 2020), MAPK12 (Bachstetter & Van Eldik, 2010), and IRAK1 (Ramasamy & Subbian, 2021) and several new host cell-specific kinases including EPHA7, MAP3K8, and MAP3K2 that are important for the release of cytokines and chemokines in myeloid cells (Fig 2C). The molecular mechanisms of how the NTD of the spike protein activates multiple kinases and how these newly identified kinases promote cytokine release in myeloid cells warrant further investigations.

Machine learning-based approaches represent a tempting opportunity for discovering new targets and drugs for human diseases. By taking advantage of our machine learning-based models, we predicted responses to 428 kinase inhibitors as single agent and 91 thousand two-drug combinations that could affect the NTD-mediated cytokine release. Our findings of several kinases involved in the cytokine release strongly suggest that simultaneous targeting of multiple host kinases involved in SARS-CoV-2-mediated cytokine production would yield in more effective treatment options than the use of more selective agents. We identified FDA-approved, multi-kinase inhibitor, ponatinib, as a potent inhibitor of cytokine production in response to the NTD from SARS-CoV-2 and its emerging variants. Despite the development of highly effective vaccines, COVID-19 will continue to be a healthcare burden, especially in persons who remain unvaccinated. We envision a potential clinical trial of ponatinib in COVID-19 patients will entail 5–7 days of treatment, similar to the baricitinib trial in severe COVID-19 patients (Bronte *et al*, 2020). This treatment period is much shorter than the ponatinib regimen given to cancer patients (median 32.1 months), reducing potential toxicity concerns. In addition, the $EC_{50}$ values of ponatinib-mediated inhibition of cytokine release in myeloid cells are in the low nano-molar ranges (< 25 nM), which is a clinically achievable dose. Overall, we believe that ponatinib and other FDA-approved drugs including cobimetinib, sunitinib, and bosutinib identified in this study could represent strong candidates for drug repurposing efforts aimed at providing an alternative and timely treatment for COVID-19 patients exhibiting major, life-threatening symptoms.

# Materials and Methods

### Reagents and Tools table

| Reagent/resource | Reference or source | Identifier or catalog number |
| --- | --- | --- |
| **PBMCs and cell lines** | | |
| THP1 | ATCC | TIB-202 |
| Raw264.7 | ATCC | TIB-71 |
| PBMCs from healthy donors | Bloodworks NW, Seattle, WA | N/A |
| PBMCs from COVID patients | The Seattle Vaccine Trials Unit | N/A |
| **Recombinant purified proteins** | | |
| Full-length S1 protein SARS-CoV-2 Wuhan | Ray Biotech Life | 230-30161 |
| RBD SARS-CoV-2 Wuhan | Ray Biotech Life | 230-30162 |
| NTD SARS-CoV-2 Wuhan | Leinco Technologies Inc | S853 |
| NTD SARS-CoV-2 Wuhan | Acro Biosystems | S1D-C52H6 |
| NTD SARS-CoV-2 U.K. (B.1.1.7) | Acro Biosystems | S1D-C52Hd |
| NTD SARS-CoV-2 South Africa (B.1.351) | Acro Biosystems | S1D-C52Hc |
| NTD SARS-CoV-2 Delta (B.1.651.2) | Acro Biosystems | S1D-C52Hf |
| S1 protein HCoV-229E | Sino Biological Inc. | 40605-V08B |
| S1 protein HCoV-OC43 | Sino Biological Inc. | 40607-V08H1 |
| **Chemicals and biologics** | | |
| Ponatinib | Selleck Chemicals | S1490 |
| Dexamethasone | Selleck Chemicals | S1322 |
| Baricitinib | Selleck Chemicals | S2851 |
| SB218078 | Tocris Bioscience | 2560 |
| Cobimetinib | Cayman Chemicals | 19563 |
| AT9283 | Selleck Chemicals | S1134 |
| Sunitinib | Cayman Chemicals | 13159 |
| Bosutinib | Tocris Bioscience | 4361 |
| TAE-684 | Cayman Chemicals | 17670 |

**Reagents and Tools table** (continued)

| Reagent/resource | Reference or source | Identifier or catalog number |
|---|---|---|
| GSK-650394 | NCGC00250410 | NCATS (NCGC) |
| Axitinib | Selleck Chemicals | S1005 |
| Phorbol 12-myristate 13-acetate | LC Laboratories | P-1680 |
| Lipopolysaccharides | Sigma Aldrich | L2630 |
| **Cell culture reagents** | | |
| RPMI1640 | Fisher Scientific | 11875135 |
| DMEM | Fisher Scientific | MT10017CM |
| FBS | Gibco | 26140-079 |
| Penn Strep. | Fisher Scientific | 15140163 |
| Sodium pyruvate | Fisher Scientific | BW13115E |
| **Luminex capture antibodies** | | |
| Mouse GMCSF | BioLegend | 505408 |
| Mouse IL-6 | BD Biosciences | 554398 |
| Mouse TNFα | BioLegend | 510804 |
| Human IL1 β | R&D Systems | MAB601 |
| Human IL-6 | R&D Systems | MAB206 |
| Human IL-8 | Fisher Scientific | M801 |
| Human IL-10 | Fisher Scientific | M010 |
| Human MCP1 | BD Biosciences | 555055 |
| Human GMCSF | BD Biosciences | 554502 |
| Human TNFα | BD Biosciences | 551220 |
| **Luminex detection antibodies** | | |
| Mouse GMCSF | BioLegend | 505502 |
| Mouse IL-6 | BioLegend | 504602 |
| Mouse TNFα | BioLegend | 506312 |
| Human IL1 β | R&D Systems | BAF201 |
| Human IL-6 | R&D Systems | BAF206 |
| Human IL-8 | Fisher Scientific | M802B |
| Human IL-10 | BioLegend | 501502 |
| Human MCP1 | BioLegend | 502609 |
| Human GMCSF | BD Biosciences | 554505 |
| Human TNFα | BioLegend | 502904 |
| **AIM assay reagents** | | |
| Human Serum AB | Gemini Bio | 100-512 |
| Cytomegalovirus control peptide pool | AnaSpec | AS-62339 |
| *Staphylococcal* enterotoxin B | Calbiochem | 11100-45-1 |
| Live/Dead Aqua | Invitrogen | L34957 |
| Cell Staining Buffer | BioLegend | 420201 |
| Human TruStain FcX | BioLegend | 422302 |
| Brilliant Stain Buffer | Fisher Scientific | BDB563794 |
| Paraformaldehyde | Fisher Scientific | 50-980-487 |
| **AIM assay antibodies** | | |
| CD40 | Miltenyi | 130-094-133 |
| CD3 ECD | Beckman Coulter | IM270SU |
| CD4 Alexa Fluor 488 | BD Biosciences | 557695 |

**Reagents and Tools table**   (continued)

| Reagent/resource | Reference or source | Identifier or catalog number |
|---|---|---|
| CD8 PerCP eFluor 710 | eBioscience | 46-0087-42 |
| CD69 BV421 | BD Biosciences | 562884 |
| CD137 APC | BD Biosciences | 550890 |
| CD154 PE | BD Biosciences | 555700 |
| **Small interfering RNA** | | |
| si-MAPK12 | Dharmacon | L-003590-00 |
| si-EPHA7 | Dharmacon | L-003119-00 |
| si-MAP3K8 | Dharmacon | L-003511-00 |
| si-PRKACG | Dharmacon | L-004651-00 |
| si-IRAK1 | Dharmacon | L-004760-00 |
| si-MAP3K3 | Dharmacon | L-003301-00 |
| si-JAK1 | Dharmacon | L-003145-00 |
| si-PTK2B | Dharmacon | L-003165-00 |
| si-MAPK14 | Dharmacon | L-003512-00 |
| si-PTK5 | Dharmacon | L-003139-00 |
| si-MAP4K2 | Dharmacon | L-003587-00 |
| si-STK17A | Dharmacon | L-005377-00 |
| si-CAMKK2 | Dharmacon | L-004842-00 |
| si-Scrambled | Dharmacon | D-001810-10 |
| **Oligonucleotides** | | |
| qPCR primers | Bio-Rad | |
| Real time primers | See Dataset EV2 | |
| **Animals** | | |
| C57BL/6J mice, male | The Jackson Laboratory | 000664 |
| **Other reagents** | | |
| CellTiter-Glo 2.0 Cell Viability Assay | Promega | G9241 |
| Lipofectamine RNAiMax | Thermo Scientific | 13778150 |
| RNeasy Mini Kit | QIAGEN | 74104 |
| RT2 First-Strand Kit | QIAGEN | 330411 |
| SYBR Green Supermix | Bio-Rad | 1725274 |
| **Instrument** | | |
| Luminex 200 instrument | Luminex | N/A |
| Plate reader | N/A | N/A |
| Bio-Rad CFX384 thermocycler | Bio-Rad | 1855484 |
| Northern Lights spectral flow cytometer | Cytek | N/A |
| **Software** | | |
| GraphPad Prism 8.0 | https://www.graphpad.com/ | N/A |
| SpectroFlo | Cytek | N/A |

## Methods and Protocols

### Cell culture

- Peripheral blood mononuclear cells (PBMCs) from healthy donors spanning various age groups and THP1 cells were cultured in RPMI1640 media supplemented with 10% FBS, 1% P/S, 1 mM sodium pyruvate.

- Raw264.7 cells were maintained in Dulbecco's minimum essential medium (DMEM) supplemented with 10% FBS and 1% Penn Strep.
- All cell lines were grown at 37°C under 5% $CO_2$, 95% ambient atmosphere.
- THP1 cells were differentiated into macrophages by inducing with phorbol 12-myristate 13-acetate (PMA) at 25 ng/ml for 24 h.

### Human peripheral blood mononuclear cells from COVID-19 patients

Consenting SARS-CoV-2-infected ($n = 19$) donors, age 18 years and older, provided anticoagulated blood samples by venipuncture at the Seattle Vaccine Trials Unit. SARS-CoV-2 donors were diagnosed by PCR testing of nasopharyngeal swabs, had mild-moderate disease, and were sampled post-symptom onset. PBMC were isolated and cryopreserved within 4 h of collection. Cell viabilities were assessed post-thawing and after 24 h of treatment. Fred Hutchinson Cancer Research Center Institutional Review Board approved all aspects of this study (IRB 10440, 00001080, and 00022371). Informed consent was obtained from all subjects, and experiments conformed to the principles set out in the WMA Declaration of Helsinki and the Department of Health and Human Services Belmont Report.

### Cytokine measurement

Cytokines were measured by Luminex multiplex assay.

- Samples and cytokine standards were incubated with Luminex microbeads (one unique bead population per cytokine) coated with cytokine-specific antibodies.
- Beads are washed then incubated 1 h with biotinylated anti-cytokine antibodies and washed again then incubated 30 min with a phycoerythrin-streptavidin conjugate.
- After a final wash, the assay is read on a Luminex 200 instrument, classifying each bead as to its cytokine-specificity and phycoerythrin fluorescence intensity. Phycoerythrin fluorescence of each bead will be proportional to the cytokine concentration in the samples or standards.
- A 5-parameter logistic standard curve is generated for each cytokine, with sample concentrations calculated from these curves.

### Kinase inhibitor screening

Kinase inhibitor screening was performed as described previously (Gujral et al, 2014a). Briefly, 35 kinase inhibitors were tested for the effect on NTD-mediated cytokine release in PBMC. All inhibitors were tested at 500 nM. Pooled PBMC from several donors were plated in 12-well plate ($1 \times 10^6$ cell per well in 1ml). Kinase inhibitors or DMSO control was subsequently added to each well. Conditioned medium collected 24 h post-treatment was snap-frozen for cytokine analysis.

### Cell viability

The effects of inhibitors on viability of PBMCs were measured using CellTiter-Glo assay as described previously (Gujral et al, 2014c; Vijay & Gujral, 2020). Briefly, cells ($5 \times 10^3$ in 100 µl culture medium) were seeded on a 96-well plate (Corning, NY, USA). Cells were then treated with various inhibitors at 500 nM as single agent. After 24 h, cells were incubated with CTG2.0 reagent for 5 min and total viability was measured by obtaining luminescent signal intensity. The quantified data were normalized to untreated control and plotted in Prism (GraphPad software).

### Small interfering RNA transfection

siRNA transfections were done in 12-well plates using Lipofectamine RNAiMax according to manufacturer instructions.

### RNA extraction and quantitative PCR

Total cellular RNA was isolated using an RNeasy Mini Kit. mRNA expression changes in genes encoding for various cytokines were determined by quantitative real-time PCR (qPCR). Briefly, 0.5–1 µg of total RNA was reverse transcribed into first-strand cDNA using an RT2 First-Strand Kit. The resultant cDNA was subjected to qPCR using human cytokine-specific primer and GAPDH as a control. The thermocycle profile used an initial denaturation step of 10 min at 95°C, followed by 15 s at 95°C and 60 s at 58°C for 40 cycles and was performed with a Bio-Rad CFX384 thermocycler. The mRNA levels of genes encoding cytokine expression were normalized relative to the mean levels of the housekeeping gene and compared using the $2-\Delta\Delta Ct$ method as described previously (Gujral et al, 2014a).

### Kinase inhibitor Regularization (KiR) modeling

KiR models for NTD-mediated release of each cytokine in PBMCs were generated as previously described (Gujral et al, 2014a). Briefly, a panel of 427 kinase inhibitors previously had their pairwise effects on 298 human kinases profiled (Anastassiadis et al, 2011; preprint: Rata et al, 2020). The result is a quantitative drug-target matrix, where each entry is a percentage between 0 and 100 that represents that kinases residual activity (as a percent of control, uninhibited activity) in the presence of that inhibitor. A set of 35 inhibitors were tested on pooled PBMCs as described above, with the end result being a single response for each drug that represents the change in cytokine release (as % DMSO control) at the profiled dose of the inhibitor (usually 500 nM). The kinase inhibition profiles of each inhibitor and the quantitative responses to those inhibitors were used as the explanatory and response variables, respectively, for elastic net regularized multiple linear regression models (Zou & Hastie, 2005). Custom R scripts (available at https://github.com/FredHutch/KiRNet-Public) employing the glmnet package were used to generate the final models (Friedman et al, 2010). Leave-one-out cross-validation (LOOCV) was used to select the optimal value for the penalty scaling factor λ. Models were computed for 11 evenly spaced values of α (the relative weighting between LASSO and Ridge regularization) ranging from 0 to 1.0 inclusive. Kinases with positive coefficients in at least one of these models (with the exception of $\alpha = 0$, which always has non-zero coefficients for every kinase) were considered hits (Figure S4A). Model accuracy was assessed via the LOOCV error as well as the root-mean-squared error of the predictions for the tested inhibitors (Figure S2).

### Deep Neural Network (DNN) Development

The development of the KiDNN models was achieved through the Keras and TensorFlow Deep Learning framework as described previously (Chollet, 2015; preprint: Abadi et al, 2016; Vijay & Gujral, 2020). Briefly, a multi-phase Grid Search method was used to optimize the DNN hyperparameters (epochs, batch size, optimizer, weight initializer, activation function, hidden layer quantity, and nodes per hidden layer) (Bergstra et al, 2011). Grid Search is a commonly employed method of hyperparameter optimization that evaluates combinations of numerous hyperparameter values to identify the model characteristics resulting in the lowest error between observed and predicted migration. The error function that was used to compare numerous models was LOOCV (leave-one-out-cross-validation) MSE (Zhang, 1993). In LOOCV, each time $n - 1$

drugs' activity profiles are used to train the model to predict the remaining drug's effect on cell migration. The process is repeated $n$ times, excluding and predicting each and every drug. Mean squared error (MSE) between predicted and observed migration is used to assign an error score to each model built with various combinations of hyperparameter values. In each phase of Grid Search, various combinations of hyperparameters are tested and the combination with the lowest LOOCV MSE is used in the subsequent phase of optimization until the final phase is reached. After optimization, the top performing hyperparameters are used to build the final KiDNN models.

### Prediction of naïve drugs and drug combinations

To predict the effect of all 428 drugs of the original matrix, the final KiDNN-cytokine models were trained on the training dataset. Keeping the weights and biases of the KiDNN-cytokine network constant, new kinase activity data for naïve drugs were inputted into the model for prediction. Further, the effect of combinations of drugs was also predicted by creating pseudo-activity matrices. Assuming two activity matrices for two different drugs defined by $[J_1, J_2, J_3, \ldots J_n]$ and $[K_1, K_2, K_3, \ldots K_n]$, where $n$ corresponds to a specific kinase's inhibition, a linear combination of the two activities corresponding to each drug for each kinase was applied to create a pseudo-activity matrix ($P$) combining both drug's effect on a kinase:

$$P_n = \frac{J_n K_n}{J_n + K_n - J_n K_n} \tag{1}$$

Subsequently, the pseudo-matrix for all 428 by 428 (including control) combinations of drugs was computed and inputted into KiDNN for prediction.

### In vitro Activation-Induced Marker (AIM) assay to assess T-cell response

A modified flow cytometric AIM assay (Reiss *et al*, 2017) was used to assess the effect of antigen-specific T cells in the presence of drug candidates.

- Single cryopreserved PBMC vials were thawed (37°C) and washed (centrifuged at 300 ×$g$) twice, resuspended in 5 ml RPMI-HEPES (supplemented with L-glutamine and penicillin/streptomycin) containing 10% Human Serum AB (R10 HS) and incubated at 37°C/5% $CO_2$ for 3 h at a concentration of $2 \times 10^7$ cells/ml.
- Cells were counted using a Guava Easycyte (Luminex) flow cytometer, resuspended in R10 HS containing 1 µg/ml anti-CD40 antibody at a concentration of $1 \times 10^7$ cells/ml. Cell suspensions (0.1 ml) for each stimulation condition were transferred to individual wells of a 96-well microtiter culture plate (Thermo Fisher) and incubated at 37°C/5% $CO_2$ for 30 min.
- Cytomegalovirus (CMV) control peptide pool, containing defined HLA-restricted epitopic synthetic peptides, was used as a positive control, and peptide pool diluent (PBS containing 0.01% DMSO) was used as a negative control.
- Ponatinib, baricitinib, or dexamethasone was serially diluted (2-fold) and added to appropriate wells at a final concentration range of 3.9–250 nM. CMV or peptide pool diluent was added to positive controls and negative controls (at each drug dilution); the final CMV peptide pool concentration was 2 µg/ml for each condition.

- Polychromatic flow cytometric (PFC) compensation controls (Live/Dead Aqua; CD3 ECD; CD4 Alexa Fluor 488, CD8 PerCP eFluor 710, CD69 BV421, CD137 APC, CD154 PE-BD) were set up in parallel without drug treatment. For CD69, CD137, and CD154, compensation control *Staphylococcal* enterotoxin B was added to a concentration of 1 µg/ml.
- Microtiter plates containing cells were centrifuged at 300 ×$g$ for 5 min. Stimulations and compensation controls were incubated at 37°C/5% $CO_2$ for 16 h.
- Cells were washed twice with 1X PBS (Thermo Fisher).
- The Live/Dead Aqua compensation control and stimulations were stained with Live/Dead Aqua for 30 min.
- Cells were washed with Cell Staining Buffer (CSB) supplemented with Human TruStain FcX (FcX CSB).
- An antibody cocktail (0.1 ml per stimulation) was prepared in FcX CSB using PFC antibodies and Brilliant Stain Buffer (BSB). Appropriate compensation controls and stimulations were stained with single antibodies (prepared in BSB) and antibody cocktail, respectively, for 30 min.
- Cells were washed with CSB and fixed with 4% paraformaldehyde in PBS for 30 min. Cells were washed again with CSB prior to flow cytometric acquisition.
- A Northern Lights spectral flow cytometer was used to acquire spectral flow cytometric data. SpectroFlo software was used to unmix and analyze data.

### Animals and experimental procedure

The study was conducted under the approval of the Institutional Care and Use Committee (IACUC) for Fred Hutchison Cancer Research Center. Male adult C57BL/6J mice (12-weeks-old, body weight 25–30 g) were used. Mice were treated with lipopolysaccharides (LPS) using oropharyngeal administration to induce lung inflammation as previously described (Allen, 2014).

- Briefly, mice were pre-treated with ponatinib at 35 mg/kg or ponatinib vehicle control (25 mM citrate buffer, pH 2.75) for 30 min via intraperitoneal injection.
- Mice were then anesthetized by isoflurane.
- Anesthetized mouse was suspended on the intubation stand by its front incisors. The tongue was gently pulled out of the mouth with the angled forceps until slight resistance was felt to access the trachea.
- While holding the tongue in the extended position, a dose of 20 µg LPS in 50 µl 1× phosphate-buffered saline (PBS) was administered into the back of the throat.
- The nostrils were covered with a gloved finger while the tongue was held extended for 5–10 additional breaths for LPS to be inhaled.
- The mouse was removed from the intubation stand and placed in a cage.
- Five hours after LPS inoculation, mice were sacrificed and bronchoalveolar lavage fluid (BALF) was collected.
- Briefly, a nick was made in the trachea. A 20-G blunt needle was inserted into the trachea, and a 1-ml syringe containing 1 ml PBS was attached to the blunt needle. The entire volume was slowly discharged to inflate the lungs and then withdrawn by pulling back the plunger to collect BALF.

- Cytokines in BALF including GMCSF, IL-6, and TNFα were measured by Luminex multiplex assay as described above.
- The lungs were formalin-fixed, embedded in paraffin blocks, and cut into 4-μm sections for hematoxylin and eosin staining.

## Data availability

The custom python scripts that implements KiDNN framework are available on GitHub: https://github.com/gujrallab/Covid-19. This repository also includes all associated files needed to execute the script and produce a sample model using the training dataset. R scripts for KiR modeling is available at https://github.com/FredHutch/KiRNet-Public

**Expanded View** for this article is available online.

## Acknowledgements

This work was supported by grants from the Fred Hutch COVID-19 Pilot Fund and Scientific Computing Infrastructure at Fred Hutch funded by ORIP grant S10OD028685. We thank Drs. Josh Schiffer, Thomas Uldrick, and Rachel Bender Ignacio for helpful discussion on clinical implications of our findings, Dr. Milka Kostic for her helpful comments and suggestions for the manuscript. We thank Dr. Thomas Bello and Nithisha Khasnavis for help with the KiR modeling, Jessica Cheney, and Julie Czartoski for help with the PBMCs processing. We also thank the Richard Lawler (Immune Monitoring Core) for his help and support with Luminex assays.

## Author contributions

MC, ECH, and TSG conceived the study. MC and TSG performed all the experiments. JM performed the AIM assay. SV performed the KiDNN modeling. MC, JM, MJM, ECH, and TSG assisted in designing, analyzing, and interpreting experiments. All authors discussed the results, and MC, SV, ECH, and TSG wrote the manuscript with comments from all authors.

## Conflict of interest
The authors declare that they have no conflict of interest.

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
