## [Review Process File · Molecular Systems Biology]

Machine Learning Identifies Molecular Regulators and Inhibitors for COVID19-induced Cytokine Release

Marina Chan, Siddharth Vijay, John McNevin, Juliana McElrath, Eric Holland, and Taran Gujral
DOI: [10.15252/msb.202110426](https://doi.org/10.15252/msb.202110426)

Corresponding author(s): Taran Gujral (tgujral@fredhutch.org)

Review Timeline:

Submission Date:	6th May 21
Editorial Decision:	11th Jun 21
Appeal Received:	22nd Jun 21
Editorial Decision:	22nd Jun 21
Revision Received:	19th Jul 21
Editorial Decision:	3rd Aug 21
Revision Received:	6th Aug 21
Accepted:	9th Aug 21

Editor: Maria Polychronidou

Transaction Report:

Thank you again for submitting your work to Molecular Systems Biology. We have now heard back from the two referees who agreed to evaluate your study. As you will see below, the reviewers raise substantial concerns on your work, which unfortunately preclude its publication in Molecular Systems Biology.

The reviewers appreciate that the goal of the study is relevant. While reviewer #2, who is a computational biologist, is somewhat more supportive, reviewer #1, who is a systems immunologist, raises significant concerns regarding the in vivo and clinical relevance of the work. Specifically, they mention that the presented findings are rather focused on the S1 response and may not reflect the cytokine storm in disease conditions. They also raise several important concerns regarding the identified drug, including the fact that the concentrations used are much higher than those clinically relevant and that it remains unclear if the drug would work in vivo. The latter concern is shared by both reviewers. Overall, the reviewers do not support publication in Molecular Systems Biology.

Taken together and given the low level of enthusiasm expressed by the reviewers, I am afraid I see no other choice than to return the manuscript with the message that we cannot offer to publish it. I am sorry that the review of your work did not result in a more favorable outcome on this occasion, but I hope that you will not be discouraged from submitting future work to Molecular Systems Biology. In any case, thank you for the opportunity to examine this work.

REFeree REPORTS

Reviewer #1:

Chan et al. report on the identification of a potentially clinically useful inhibitor to block cytokine production induced by the S1 protein. The authors also identify the n-terminal domain as the key S1 protein portion stimulating cytokine production in the monocytic THP1 cell line or in human PBMC. The authors utilized their previously published machine learning method for arriving at the identified drug target, which quite cleverly allows them to use a smaller inhibitor screen to identify potential drugs from the larger database of compounds. Identifying clinical useful inhibitors that can block Sars-CoV-2 induced cytokine release is of course of great interest and importance during the ongoing pandemic. Overall, I find the data clearly presented and the manuscript well written. A main limitation of the study is that the evidence for the utility of the drug is confined to the

response to S1 protein. The title should be revised to say something like 'Sars-CoV-2 S1 protein-induced cytokine release' rather than 'cytokine storm', since the latter implies the virus-infection induced phenotype within patients; I would change the reference to cytokine storm to 'cytokine production' in the text as well. As mentioned below, S1 may not be the most relevant mechanism for detrimental excess cytokine production in vivo during Sars-CoV-2 infection. Another problem is that the identified drug might be a non-starter for clinical trials due to its known adverse side effect in inducing blood clots, which is also a potential problem in severe COVID-19.

I would also like the authors to address whether the inhibition they see is achievable at clinically relevant concentrations. The maximum dose of ponatinib used in the cell culture used in this study looks to be about tenfold higher than the best info I could find on in vivo concentrations at the therapeutic doses for cancer

(https://www.accessdata.fda.gov/drugsatfda_docs/nda/2012/203469orig1s000clinpharmr.pdf). The authors do show ponatinib to still have some inhibitory efficacy at lower concentrations, but this seems to be variable between the different experiments shown in figure 3b and EV5.

Finally, although the Ponatinib finding seems to occur very prominently in the dual drug predictions, the authors never mention what drugs it may be combined with, or test the combinations experimentally.

Major points:

- Figure 4: I am not sure about the relevance of using cells from patients with COVID-19 in this way. It would be more relevant to look at cells from the bronchoalveolar lavage and determine if Ponatinib blocks their spontaneous (presumably Sars-CoV-2 stimulated) cytokine production, but of course that is very challenging to do in this situation. What is the hypothesis of why these samples would be different from the healthy donor PBMCs already shown, in terms of the response to S1 in vitro? Currently, as presented I think it is misleading as it appears to be a validation that this works in COVID-19, so this at a minimum needs to be further clarified.
- S1 NTD is not necessarily the key factor stimulating cytokine release in COVID-19; direct sensing of the virus by TLRs is likely involved. I believe the authors should at least demonstrate if Ponatinib also blocks cytokine induction various TLR stimulation, if they cannot use whole viral particles (which would be preferred). Exactly what is stimulating the cytokine storm in humans is unclear, which makes this difficult to assess, and the authors should be explicit that the S1 results do not necessarily extend to the true 'cytokine storm' as is present in the patients. Also, in response to S1, viral particles, or TLR ligands, it would also be important to assess the effect of the drug on interferon production. Furthermore, it should be noted that not all the viral-stimulated proteins that may be inhibited by ponatinib may be detrimental in the disease, in fact some immune factors important for resolution and viral clearance could be blocked and actually worsen disease.
- The dual drug predictions in the supplemental table should list the names of the drugs in the combination.

Minor Points:

- Ponatinib may have side effect issues with inducing blood clots/platelet activation, which could be a problem in its use in severe COVID-19. Can the authors identify and validate any other candidates from their study? I think more broadly highlighting several candidates could be more useful in stimulating follow up studies and trials.
- Code deposited in github could have a more detailed readme and more extensive commenting in the scripts to better allow reuse/exploration.
- Figure 2: It would be interesting to have a random 'background' set of kinases that were not chosen or lower ranked by the method, to see how often siRNA inhibition of those would diminish cytokine levels. This would verify that the selection procedure is yielding better hits than randomly chosen targets. I don't consider the addition of this to be necessary for publication (the finding of the inhibitor is the most important thing rather than the selection method), but it could help guide others that might try this approach in other contexts.

Reviewer #2:

In this study, the authors utilized a few machine learning methods to identify drugs to inhibit the cytokine storms induced by the CoV-SARS-2 spike protein. The top prediction Ponatinib is validated in-vitro with cell culture models. In general, the paper is clearly written, and the topic is essential given the COVID-19 pandemic. However, I have a few concerns for the authors to consider before publication.

1, please explain the difference between KiDNN and KiR in the main text. The authors applied two machine learning methods to prioritize kinase inhibitors. Please include the intuition behind each method in the main text. Although the model training procedure is included, the general idea behind each approach is still lacking. Also, please explain the difference between the two methods and why both are necessary for kinase inhibitor prioritization.

2, lack of intuitive explanations for the drug combination prediction. In the paper, the authors stated that "we predicted responses to 91 thousand two-drug combinations, as well as responses to 13 million combinations involving three-drug cocktails." Please include an intuitive explanation of drug combination prediction in the main text so readers can get a general idea. Especially, please clarify the format of training data and the scoring function for drug combinations.

3, lack of in-vivo evidence of Ponatinib's effect on cytokine storm. The authors have validated the Ponatinib in cell culture models treated with NTD. However, a critical issue will be the actual effect of Ponatinib upon CoV-SAR2-2 infection. If other parts of the virus can also induce cytokine storm independently of NTD, the impact of Ponatinib will only be partial. Also, the in-vivo behavior in murine models of any drugs will be critical for its clinical efficacy evaluation. If performing such an experiment is challenging for authors under ABSL-3, the authors may consider evaluating Ponatinib's therapeutic effect under ABSL-2 using mouse coronavirus MHV as a surrogate virus.

Note: if there is time pressure to publish this work, authors and editors may ignore this concern.

** As a service to authors, EMBO Press offers the possibility to directly transfer declined manuscripts to another EMBO Press title or to the open access journal Life Science Alliance launched in partnership between EMBO Press, Rockefeller University Press and Cold Spring Harbor Laboratory Press. The full manuscript and if applicable, reviewers' reports, are automatically sent to the receiving journal to allow for fast handling and a prompt decision on your manuscript. For more details of this service, and to transfer your manuscript please click on Link Not Available. **

Reviewer #1:

We are grateful to the reviewer for his thoughtful suggestions and comments. We have already generated most of the data requested by the reviewer and would be included in the revised manuscript.

Major points

1. the identified drug might be a non-starter for clinical trials due to its known adverse side effect in inducing blood clots, which is also a potential problem in severe COVID-19.

I would also like the authors to address whether the inhibition they see is achievable at clinically relevant concentrations. The maximum dose of ponatinib used in the cell culture used in this study looks to be about tenfold higher than the best info I could find on in vivo concentrations at the therapeutic doses for cancer

(https://www.accessdata.fda.gov/drugsatfda_docs/nda/2012/203469orig1s000clinpharmr.pdf). The authors do show ponatinib to still have some inhibitory efficacy at lower concentrations, but this seems to be variable between the different experiments shown in figure 3b and EV5.

We appreciate the reviewer for this safety-related concern. The IC_{50} values of Ponatinib-mediated inhibition of cytokine release in monocytes are in the low nanomolar ranges (1nM-27nM, **Fig 3b**). In consultation with Takeda Pharmaceuticals (manufacturer of Ponatinib), we have determined that this dose is clinically achievable. Notably, the clinical trial of Ponatinib in COVID19 patients would be 5-7 days of treatment which is much shorter than the time ponatinib is given to cancer patients (median 32.1 months). Further, cancer patients treated with ponatinib are heavily pre-treated, elderly and have co-morbidities. Specifically, the median onset of arterial occlusion observed in cancer patients occur at 13.4 months. No significant adverse effects of ponatinib were observed in the first two weeks. Overall, we are confident that our clinical strategy mitigates ponatinib's potential adverse effect and designed a Phase IIa/b trial to assess whether a 7-day course of ponatinib at 15 mg orally is safe and well-tolerated in outpatients recently diagnosed with mild to moderate SARS CoV2. As suggested by the reviewer, we will include the discussion of these points in the revised manuscript.

2. Finally, although the Ponatinib finding seems to occur very prominently in the dual drug predictions, the authors never mention what drugs it may be combined with, or test the combinations experimentally.

Ponatinib, as a single agent, potently inhibited S1-mediated cytokine production in monocytes. The addition of other drugs with ponatinib marginally improves overall response. Since Ponatinib is FDA-approved, we chose to focus our studies on Ponatinib as a single agent. At the request of the reviewer, we will provide additional experimental testing of drug combinations.

3. Figure 4: I am not sure about the relevance of using cells from patients with COVID-19 in this way. It would be more relevant to look at cells from the bronchoalveolar lavage and determine if Ponatinib blocks their spontaneous (presumably Sars-CoV-2 stimulated) cytokine production, but of course that is very challenging to do in this situation. What is the hypothesis of why these samples would

be different from the healthy donor PBMCs already shown, in terms of the response to S1 *in vitro*? Currently, as presented I think it is misleading as it appears to be a validation that this works in COVID-19, so this at a minimum needs to be further clarified.

In the revised manuscript, we will clarify these points raised by the reviewer. This is something we have also considered to validate; unfortunately, due to the safety requirements of ABSL-3+ standard, we are unable to perform direct testing of Ponatinib on SARS-CoV-2-mediated cytokine storm in bronchoalveolar lavage fluid (BALF) from COVID19 patients. However, we have shown that Ponatinib inhibits cytokine production in BALF from lipopolysaccharide(LPS)-induced mouse model of lung inflammation. In the revised manuscript, we will include the following result section and the figure.

Ponatinib alleviates symptoms of acute lung inflammation in the LPS-induced lung inflammation mouse model. The LPS-induced models of Acute lung injury (ALI) and acute respiratory distress syndrome (ARDS) in mice are well-established *in vivo* models to study pulmonary infection. Further, both ALI and ARDS are known to occur in the clinical presentation of severe SARS-CoV-2 disease. We show that a 20-minute pre-treatment with Ponatinib (35mg/Kg) significantly reduces symptoms of acute lung inflammation in the LPS-induced lung inflammation mouse model(**Fig. S1A**). BALF collected at 5 hours post LPS treatment showed a significant reduction in GMCSF, IL-6, and TNF α levels measured by Luminex(**Fig. S1B**). Consistently, we observed substantial suppression of cytokine expression in lung homogenates measured by qPCR (**Fig. S1B**). Together, these data suggest that Ponatinib can inhibit

Fig. S1. Ponatinib alleviates symptoms of acute lung inflammation in LPS-induced lung inflammation mouse model. (A) Representative H&E images showing ponatinib alleviates LPS induced intense septal inflammatory infiltration of polymorphonuclear cells, septal thickening and collapse as well as irregular distribution of air spaces in mouse lungs. (B) Pre-treatment (20min) with Ponatinib (35mg/Kg) inhibits LPS -induced cytokine release in the BALF fluid and (C) lung homogenates.

pulmonary inflammation in an ARDS/ALI induced by the LPS model.

4. S1 NTD is not necessarily the key factor stimulating cytokine release in COVID-19; direct sensing of the virus by TLRs is likely involved. I believe the authors should at least demonstrate if Ponatinib also blocks cytokine induction various TLR stimulation, if they cannot use whole viral particles (which would be preferred).

We thank the reviewer for this suggestion. During the peer-review process, a study published in Immunity, showed that SARS-Cov-2 induces cytokine production in human myeloid cells. This study replicated some of the findings in our manuscript, including the demonstration that NTD of S1 spike protein induces cytokine production. They also discovered new monocyte-specific receptors that recognize S1 spike protein to promote cytokine production. These new findings support our data and highlight an ACE2-independent mechanism of immune cell regulation in COVID19. In the revised manuscript, we will discuss these new data.

To address reviewer's suggestion, we've already evaluated the efficacy of ponatinib in the lipopolysaccharide(LPS)-induced ex vivo and in vivo models (**Fig S1 above**). Since the focus of our manuscript is on SARS-CoV-2 S1 protein-mediated cytokine storm, the LPS data were not included. As suggested by the reviewer, we will have the following data in the revised manuscript:

Ponatinib inhibits LPS-mediated cytokine expression in THP1 monocytes. We sought to compare S1 protein and LPS-mediated changes in cytokine expression in THP1 monocytes. Our data show that both S1 spike protein (1 μ g/mL) and LPS (1 μ g/mL) stimulation increased the expression of all measured cytokines (**Fig. S2A**). However, some individual differences in cytokine expression were also observed, such as LPS stimulation caused ~1000-fold increase in the expression of IL-6 compared with 10-fold increase caused by S1 spike protein stimulation (**Fig. S2A**). Conversely, S1 protein stimulation caused ~1000-fold change in the expression of CXCL10 compared with a 10-fold increase caused by LPS stimulation. Importantly, ponatinib treatment inhibited expression of LPS-mediated expression of IL-1b, IL-6, IL-8, and TNFa(**Fig. S2B**). Consistently, our data show that 1-hour pre-treatment with ponatinib inhibits LPS-induced, the time-dependent activity of several signaling pathways, including phosphorylation of STAT3, NFkb, JNK, SRC, and p38 MAPK in THP1 cells (**Fig. S2C**). No changes in the LPS-mediated phosphorylation of ERK1/2 were observed. Together, these data suggest that ponatinib also blocks LPS-mediated downstream signaling and cytokine expression in monocytes.

Fig. S2. Ponatinib blocks LPS-mediated downstream signaling and cytokine expression. (A) Comparison of LPS and S1 spike protein-mediated changes in cytokine expression in THP1 cells. (B) Pre-treatment (60min) with Ponatinib (1 μ M) inhibits LPS -induced cytokine release in THP1 cells (C) Plots showing changes in phosphorylation of indicated proteins in response to LPS treatment. THP1 cells were pre-treated with ponatinib or DMSO control for 60 min followed by stimulation with LPS at indicated times. Changes in the phosphorylation of indicated proteins were measured using reverse-phase protein arrays. Bars indicate mean of at least two independent replicates and error bars indicate SEM.

5. Also, in response to S1, viral particles, or TLR ligands, it would also be important to assess the effect of the drug on interferon production. Furthermore, it should be noted that not all the viral-stimulated proteins that may be inhibited by ponatinib may be detrimental in the disease, in fact some immune factors important for resolution and viral clearance could be blocked and actually worsen disease.

In the revised manuscript, we will include measurements on interferon production in response to S1 protein.

Minor points

6. The dual drug predictions in the supplemental table should list the names of the drugs in the combination.

We thank the reviewer for pointing this out. In the revised manuscript, we will include the names of the drugs in the combination.

7. Ponatinib may have side effect issues with inducing blood clots/platelet activation, which could be a problem in its use in severe COVID-19. Can the authors identify and validate any other candidates from their study? I think more broadly highlighting several candidates could be more useful in stimulating follow up studies and trials.

Again, we thank the reviewer for this suggestion. We have already experimentally validated several other FDA-approved or clinical-grade inhibitors that could inhibit S1 protein-mediated cytokine release, including cobimetinib, baricitinib, and sunitinib. These data will be clearly highlighted as a separate figure in the revised manuscript.

8. Code deposited in github could have a more detailed readme and more extensive commenting in the scripts to better allow reuse/exploration.

We will provide a more detailed readme file and better commenting in the script.

9. Figure 2: It would be interesting to have a random 'background' set of kinases that were not chosen or lower ranked by the method, to see how often siRNA inhibition of those would diminish cytokine levels. This would verify that the selection procedure is yielding better hits than randomly chosen targets. I don't consider the addition of this to be necessary for publication (the finding of the inhibitor is the most important thing rather than the selection method), but it could help guide others that might try this approach in other contexts.

We have extensively characterized and validated our kinase selection method including validating 'randomly chosen' kinases in our previous publications; KiR (PNAS, 2014, Nat Comm, 2017) and KiDNN (iScience, 2020). These points will be clearly discussed in the methods and discussion section of our revised manuscript.

Reviewer #2:

1. please explain the difference between KiDNN and KiR in the main text. The authors applied two machine learning methods to prioritize kinase inhibitors. Please include the intuition behind each method in the main text. Although the model training procedure is included, the general idea behind each approach is still lacking. Also, please explain the difference between the two methods and why both are necessary for kinase inhibitor prioritization.

In the revised manuscript, we will clearly explain the difference between KiDNN (iScience, 2020) and KiR (PNAS, 2014, Nat Comm, 2017) in the Results and Methods section.

2. lack of intuitive explanations for the drug combination prediction. In the paper, the authors stated that "we predicted responses to 91 thousand two-drug combinations, as well as responses to 13 million combinations involving three-drug cocktails." Please include an intuitive explanation of drug combination prediction in the main text so readers can get a general idea. Especially, please clarify the format of training data and the scoring function for drug combinations.

We thank the reviewer for pointing this out. An explanation of how the drug combination was computed is included in the Supplementary Methods section. In the revised manuscript, we will clearly include this section in the main text.

3. lack of in-vivo evidence of Ponatinib's effect on cytokine storm. The authors have validated the Ponatinib in cell culture models treated with NTD. However, a critical issue will be the actual effect of Ponatinib upon CoV-SAR2-2 infection. If other parts of the virus can also induce cytokine storm independently of NTD, the impact of Ponatinib will only be partial. Also, the in-vivo behavior in murine models of any drugs will be critical for its clinical efficacy evaluation. If performing such an experiment is challenging for authors under ABSL-3, the authors may consider evaluating Ponatinib's therapeutic effect under ABSL-2 using mouse coronavirus MHV as a surrogate virus.

Note: if there is time pressure to publish this work, authors and editors may ignore this concern.

Unfortunately, due to the safety requirements of ABSL-3 standard, we are unable to perform direct testing of Ponatinib on SARS-CoV-2-mediated cytokine storm in the animal model. However, we have evaluated the efficacy of ponatinib in LPS-induced cytokine storm in mice. Please see the response to points 3 and 4 (Reviewer 1, **Fig S1, S2 above**). These data will be included in the revised manuscript.

Thank you once again for the constructive call yesterday, where we discussed re-considering our decision on your manuscript MSB-2021-10426.

As we discussed, and after reading once again your preliminary point by point response, I think that the envisioned revisions seem promising for addressing the reviewers' concerns. As such, I would invite you to submit a revised version once you have completed the additional experiments and analyses.

As we concluded during the call, it is particularly important to demonstrate more convincingly that:

- Ponatinib is a clinically relevant inhibitor of SARS-CoV2 (i.e. by showing that it works in an animal model, that it does not suppress relevant immune responses but only alleviates the detrimental cytokine storm, and by testing it in further experimental setups, including in LPS-induced inflammation models).
- The presented systems approach identifies further relevant drugs (or combinations), beyond the selected example of Ponatinib.

When you submit your revised manuscript, please attach a point by point response detailing how you have handled each of the points raised by the referees. A revised manuscript will be once again subject to review and you probably understand that we can give you no guarantee at this stage that the eventual outcome will be favorable.

Reviewer #1:

We are grateful to the reviewers for their thoughtful suggestions and comments. We will address each of their specific concerns individually

Major points

1. the identified drug might be a non-starter for clinical trials due to its known adverse side effect in inducing blood clots, which is also a potential problem in severe COVID-19.
I would also like the authors to address whether the inhibition they see is achievable at clinically relevant concentrations. The maximum dose of ponatinib used in the cell culture used in this study looks to be about tenfold higher than the best info I could find on in vivo concentrations at the therapeutic doses for cancer (https://www.accessdata.fda.gov/drugsatfda_docs/nda/2012/203469orig1s000clinpharmr.pdf). The authors do show ponatinib to still have some inhibitory efficacy at lower concentrations, but this seems to be variable between the different experiments shown in figure 3b and EV5.

We appreciate the reviewer for this safety-related concern. We envision a potential clinical trial of Ponatinib (or other drugs identified in our screen) in COVID-19 patients will entail 5-7 days of treatment, similar to the Baricitinib trial in severe COVID-19 patients (Bronte et al, 2020). This treatment period is much shorter than the ponatinib regimen given to cancer patients (median 32.1 months), reducing potential toxicity concerns. Moreover, cancer patients treated with ponatinib are heavily pre-treated with other drugs, elderly, and comorbidities. Specifically, the median onset of arterial occlusion observed in cancer patients occur at 13.4 months. No significant adverse effects of Ponatinib were observed in the first two weeks of the clinical trials. In addition, the EC₅₀ values of Ponatinib-mediated inhibition of cytokine release in monocytes are in the low nano-molar ranges (<25nM) (**New Fig 4B, Page 26**), which is a clinically achievable dose- conclusion verified by both Clinical Oncologists at our institution and in consultation with Takeda Pharmaceuticals (manufacturer of Ponatinib). Overall, we are confident that our clinical strategy mitigates Ponatinib's potential adverse effect and designed a Phase IIa/b trial to assess whether a 7-day course at 15 mg orally is safe well-tolerated in outpatients recently diagnosed with mild to moderate SARS CoV2. As suggested by the reviewer, we have now included the above points in the discussion section of the revised manuscript (Page 11).

2. Finally, although the Ponatinib finding seems to occur very prominently in the dual drug predictions, the authors never mention what drugs it may be combined with, or test the combinations experimentally.

Ponatinib, as a single agent, potently inhibited S1-mediated cytokine production in monocytes (**New Fig 3B, 4A, 4B**). The addition of other drugs with ponatinib marginally improves overall response (see **Dataset 1**). Since Ponatinib is FDA-approved, we chose to focus our studies on Ponatinib as a single agent. However, in the revised manuscript, we also include experimental validation of additional nine inhibitors, including 5 FDA-approved inhibitors (See **Fig. 3B**)

3. Figure 4: I am not sure about the relevance of using cells from patients with COVID-19 in this way. It would be more relevant to look at cells from the bronchoalveolar lavage and determine if Ponatinib blocks their spontaneous (presumably Sars-CoV-2 stimulated) cytokine production, but of course that is very challenging to do in this situation. What is the hypothesis of why these samples would be different from the healthy donor PBMCs already shown, in terms of the response to S1 in vitro? Currently, as presented I think it is misleading as it appears to be a validation that this works in COVID-19, so this at a minimum needs to be further clarified.

In the revised manuscript, we have clarified these points raised by the reviewer (see Page 8). We did not intend our work to use COVID-19 PBMCs as a validation. Previous studies have shown that a subset of monocytes may be infected by SARS-CoV-2 and differentiate into infected macrophages in local tissue, contributing to the inflammatory cytokine storm in patients (Jafarzadeh *et al*, 2020) (Guo *et al*, 2020). Therefore, we were curious to determine whether PBMCs isolated from COVID-19 patients respond differently to NTD treatment from healthy donor PBMCs. We found that the fold change in a subset of cytokines released in response to NTD stimulation was significantly higher in COVID-19 PBMCs than healthy donor PBMCs (**New Fig. EV5**). Whether this difference can be attributed to the duration of infection, clinical presentation, or patient characteristics such as age and pre-existing conditions, warrants future investigations. Nonetheless, Ponatinib still outperforms Baricitinib and dexamethasone in inhibiting NTD stimulated cytokine release from COVID-19 PBMCs (**now moved to EV, Fig. EV5**).

Unfortunately, due to the safety requirements of ABSL-3+ standard, we are unable to perform direct testing of Ponatinib on SARS-CoV-2-mediated cytokine storm in bronchoalveolar lavage fluid (BALF) from COVID19 patients. However, in the revised manuscript, we show that Ponatinib inhibits cytokine production in BALF from lipopolysaccharide(LPS)-induced mouse model of lung inflammation (See **New Fig.5E**, Page 28) and the results section on **Page 10**.

4. S1 NTD is not necessarily the key factor stimulating cytokine release in COVID-19; direct sensing of the virus by TLRs is likely involved. I believe the authors should at least demonstrate if Ponatinib also blocks cytokine induction various TLR stimulation, if they cannot use whole viral particles (which would be preferred).

We thank the reviewer for this suggestion. During the peer-review process, a study published in Immunity, showed that SARS-Cov-2 induces cytokine production in human myeloid cells. This study replicated some of the findings in our manuscript, including demonstrating that NTD of S1 spike protein induces cytokine production. They also discovered new monocyte-specific receptors that recognize S1 spike protein to promote cytokine production. These new findings support our data and highlight an ACE2-independent mechanism of immune cell regulation in COVID-19. In the revised manuscript, this new finding is described in the Introduction section on **Page 3**.

To address the reviewer's suggestion, we evaluated the efficacy of Ponatinib in the lipopolysaccharide(LPS)-induced in vitro and in vivo models (**New Fig. 5, Page 27**). Since the focus of our manuscript is on SARS-CoV-2 S1 protein-mediated cytokine storm, the

LPS data were not included in the original submission. As suggested by the reviewer, we have incorporated these data in the revised manuscript.

Our data show that a 1-hour pre-treatment with Ponatinib (35 mg/Kg) significantly reduces symptoms of acute lung inflammation in the LPS-induced lung inflammation mouse model (**New Fig.5B, C**). Treatment with Ponatinib at 35 mg/kg alleviated LPS-induced lung injury, including interstitial and intra-alveolar edema, septal thickening, alveolar collapse, and inflammatory cell infiltration, assessed by histology. Bronchoalveolar lavage fluid (BALF) collected at 5 hours post LPS treatment showed a significant reduction in GM-CSF, IL-6, and TNF α levels measured by Luminex (**Fig. 5D**). Together, these data suggest that ponatinib also blocks LPS-mediated downstream signaling and cytokine expression in monocytes. These new results are now described on **page 10** of the revised manuscript.

5. Also, in response to S1, viral particles, or TLR ligands, it would also be important to assess the effect of the drug on interferon production. Furthermore, it should be noted that not all the viral-stimulated proteins that may be inhibited by ponatinib may be detrimental in the disease, in fact some immune factors important for resolution and viral clearance could be blocked and actually worsen disease.

In the revised manuscript, we include measurements on interferon-gamma expression in response to NTD of SARS-CoV-2 (**See Fig. EV6** and Results section on **Page 9**).

We agree with the reviewer that a functional T-cell response is critical to the resolution of viral infection. We, therefore, employed an in vitro Activation Induced Marker (AIM) assay to assess the effect of Ponatinib on T cell response at concentrations that strongly inhibit myeloid cell-induced cytokine production. In vitro AIM assays have been used to determine T cell response to viral peptides with or without drug interventions, including SARS-CoV-2 (Moderbacher et al, 2020). To determine whether ponatinib affected anti-viral T cell function at the concentrations used to block the NTD-induced cytokine release, we treated human PBMCs with synthetic cytomegalovirus (CMV) peptide pool in the presence of various concentrations of Ponatinib, dexamethasone, and Baricitinib. Treatment with Ponatinib had little or no effect on CMV peptide-induced CD8+ T cell activation at 10-40-fold above the EC50 concentrations of myeloid cell-induced cytokine production, while some inhibitory effect on CD4+ T cells was observed at high concentrations (Fig EV6). Notably, both dexamethasone and Baricitinib induced stronger or similar inhibition on CD8+ and CD4+ T cells at the same doses. In addition, Ponatinib did not affect PBMC cell viability even at 1000nM (Dataset 1). Taken together, these data show that Ponatinib is highly effective in blocking NTD-mediated cytokine production by myeloid cells while preserving the ability of a T-cell response.

The above results are described on **Pages 9** and **New Fig EV 7**.

Minor points

6. The dual drug predictions in the supplemental table should list the names of the drugs in the combination.

We thank the reviewer for pointing this out. In the revised manuscript, we have now included the names of the drugs in the combination (**See Dataset 1**).

7. Ponatinib may have side effect issues with inducing blood clots/platelet activation, which could be a problem in its use in severe COVID-19. Can the authors identify and validate any other candidates from their study? I think more broadly highlighting several candidates could be more useful in stimulating follow up studies and trials.

Again, we thank the reviewer for this suggestion. In the revised manuscript, we provide experimental validation of additional 9 inhibitors, including 5 FDA-approved inhibitors (See **Fig. 3B, Page 25**).

8. Code deposited in github could have a more detailed readme and more extensive commenting in the scripts to better allow reuse/exploration.

We provide a more detailed readme file and better commenting in the script.

9. Figure 2: It would be interesting to have a random 'background' set of kinases that were not chosen or lower ranked by the method, to see how often siRNA inhibition of those would diminish cytokine levels. This would verify that the selection procedure is yielding better hits than randomly chosen targets. I don't consider the addition of this to be necessary for publication (the finding of the inhibitor is the most important thing rather than the selection method), but it could help guide others that might try this approach in other contexts.

We have extensively characterized and validated our kinase selection method, including validating 'randomly chosen' kinases in our previous publications, KiR (PNAS, 2014, Nat Comm, 2017).

Reviewer #2:

1. please explain the difference between KiDNN and KiR in the main text. The authors applied two machine learning methods to prioritize kinase inhibitors. Please include the intuition behind each method in the main text. Although the model training procedure is included, the general idea behind each approach is still lacking. Also, please explain the difference between the two methods and why both are necessary for kinase inhibitor prioritization.

In the revised manuscript, we explain the difference between KiDNN (iScience, 2020) and KiR (PNAS, 2014, Nat Comm, 2017) in the Results section (**See Page 5**). KiR utilizes elastic net regularization to identify kinases underlying a given phenotype, such as cell growth or release of secreted factors, e.g., cytokines. Thus, we applied KiR to predict the underlying 'kinases'. In contrast, KiDNN uses Deep Neural Networks to predict the response to 'unseen' kinase inhibitors. Although both KiR and KiDNN could predict the response to inhibitors, our previous work showed that KiDNN predictions, based on the non-linear model, were more accurate than linear model-based KiR predictions (iScience, 2020). Thus, we employed KiDNN for predicting response to unseen inhibitors as single agents and in combinations.

2. lack of intuitive explanations for the drug combination prediction. In the paper, the authors stated that "we predicted responses to 91 thousand two-drug combinations, as well as responses to 13 million combinations involving three-drug cocktails." Please include an intuitive explanation of drug combination prediction in the main text so readers can get a general idea.

Especially, please clarify the format of training data and the scoring function for drug combinations.

We thank the reviewer for pointing this out. An explanation of how the drug combination was computed is included in the Methods section. In the revised manuscript, we will clearly include this section in the main text on Page 7

3. lack of in-vivo evidence of Ponatinib's effect on cytokine storm. The authors have validated the Ponatinib in cell culture models treated with NTD. However, a critical issue will be the actual effect of Ponatinib upon CoV-SAR2-2 infection. If other parts of the virus can also induce cytokine storm independently of NTD, the impact of Ponatinib will only be partial. Also, the in-vivo behavior in murine models of any drugs will be critical for its clinical efficacy evaluation. If performing such an experiment is challenging for authors under ABSL-3, the authors may consider evaluating Ponatinib's therapeutic effect under ABSL-2 using mouse coronavirus MHV as a surrogate virus.

Note: if there is time pressure to publish this work, authors and editors may ignore this concern.

Unfortunately, due to the safety requirements of the ABSL-3 standard, we are unable to perform direct testing of Ponatinib on SARS-CoV-2-mediated cytokine storm in the animal model. However, we have evaluated the efficacy of Ponatinib in LPS-induced pulmonary inflammation model in mice. Please see the response to points 3 and 4 (Reviewer 1, **New Fig. 5, New Results section, Page 10**).

Thank you for sending us your revised manuscript. We have now heard back from the two reviewers who were asked to evaluate your revised study. As you will see below, the reviewers think that the performed revisions have addressed their concerns and they are now supportive of publication. Reviewer #1 only raises two remaining issues regarding Figure EV5 and indicating the numbers of replicates in all figures. We would ask you to address these issues in a minor revision.

We would ask you to address some remaining editorial issues listed below.

REFEREE REPORTS

Reviewer #1:

The authors have addressed the comments raised in the original review. Regarding In vivo model supporting Ponatinib as a drug for COVID-19, the authors now report the inhibition of LPS-induced lung inflammation, which supports the conclusion of Ponatinib as a strong anti-inflammatory for LPS, but not necessarily for S1 protein. The reviewer acknowledges that the actual viral infection model is difficult, and this evidence is at least supportive. I could imagine other models of trans-expression or exogenous administration of S1 protein that might have tested the hypothesis in vivo more directly, but again such a model would need some extended time to set up. The LPS model also does not allow to assess whether the strong anti-inflammatory effect of Ponatinib would be detrimental during Sars-CoV-2 infection. Given current knowledge of severe COVID-19 disease course, and given evidence presented by the authors that T cell responses are preserved to a degree similar to Baricitinib, this does not seem to be a major concern, and furthermore a full pre-clinical animal study for ponatinib in COVID-19 could be considered beyond the scope of the current paper. Additional, beyond ponatinib, the authors do provide data on several other potential drugs, and better expose the rationale behind focusing on ponatinib.

Additional points: EV5: Baricitinib mentioned in text but not shown.

Figure 5: Are there only replicates from a single experiment shown, or are there multiple experiments performed reproducing these results? For all figures, information on the number or replicate experiments performed should be added.

Reviewer #2:

The authors have addressed my concerns in the previous review round, although the LPS-induced inflammation may not directly validate the effects of corona-virus infection. I understand the difficulty of ABSL-3, but suggest the authors consider mouse MHV as a surrogate under ABSL-2 in their future studies.

Reviewer #1:

We are grateful to the reviewers for their thoughtful suggestions and comments. We will address each of their specific concerns individually

1. Additional points: EV5: Baricitinib mentioned in text but not shown.

This has been corrected on Page 8.

2. Figure 5: Are there only replicates from a single experiment shown, or are there multiple experiments performed reproducing these results? For all figures, information on the number or replicate experiments performed should be added.

In Figure 5, data are the mean of five biological replicates from two independent studies. In the revised manuscript, we have included information on the number of replicates performed for all figures.

Editorial Issues

- We have separated the results and discussion sections
- The author contribution statement has been added to the main text
- The Materials and Methods section is now presented as “Structured Methods”, including a Reagents and Tools table
- Standfirst text and synopsis image are included in the revised submission

Thank you again for sending us your revised manuscript. We are now satisfied with the modifications made and I am pleased to inform you that your paper has been accepted for publication.

Corresponding Author Name: Taran Gujral

Manuscript Number: MSB-2021-10426